## Replication

behaviour/cognition/psychology

control, domain-general, replications, cognitive control, language

**Author for correspondence:**
Carolin Dudschig
e-mail: carolin.dudschig@uni-tuebingen.de

# Are control processes domain-general? A replication of 'To adapt or not to adapt? The question of domain-general cognitive control' (Kan *et al.* 2013)

## Carolin Dudschig

Fachbereich Psychologie, University Tübingen, Schleichstr. 4, 72076 Tübingen

ID CD, 0000-0002-7041-7798

Conflict and conflict adaptation are well-studied phenomena in experimental psychology. Standard tasks investigating causes and outcomes of conflict during information processing include the Stroop, the Flanker and the Simon task. Interestingly, recent research efforts have moved toward investigating whether conflict in one task domain influences information processing in another task domain, typically referred to as cross-task conflict adaptation. These transfer effects are of central importance for theories about our cognitive architecture, as they are interpreted as pointing towards domain-general cognitive mechanisms. Given the importance of these cross-task transfer effects, the current paper targets at replicating one of the key findings. Specifically, Kan *et al.* (Kan *et al.* 2013 *Cognition* **129**, 637–651) showed that reading syntactically ambiguous sentences result in processing adjustments in subsequent Stroop trials. This result is in line with the idea that conflict monitoring works in a domain overarching manner. The present paper presents two replication studies: (i) exact replication: identical sentence-reading task intermixed with stimulus-based Stroop task and (ii) conceptual replication: identical sentence-reading task intermixed with response-based Stroop task. Power calculations were based on the original paper. Both experiments were pre-registered. Despite the experiments being closely designed according to the original study, there was no evidence supporting the hypothesis regarding cross-domain conflict adaptation.

# 1. Introduction

The interconnection between language and non-linguistic cognition is probably among the most intriguing questions within cognitive science, especially as it points to the evolution of human language. 'Is language a new machine built out of old parts?' [1,2], or, alternatively, is the human language system a fully independent module with minimal connections to non-linguistic cognition? Whereas language in the traditional view is often regarded as a high-level cognitive process that acts independently from other non-linguistic modules, there are also theories supporting a strong interconnection between linguistic and non-linguistic systems. Such theories can be summarized under the terms: grounded cognition or embodiment.

The question regarding the interconnection between the linguistic system and non-linguistic cognition has been addressed mainly on a representational level, with research investigating whether non-linguistic sensorimotor brain areas are also active during language comprehension. Additionally, recent studies show that specific processing mechanisms might transfer between linguistic and non-linguistic cognition. Specifically, the detection of processing conflict in one domain—for example, in the linguistic domain—might result in conflict adjustments in a non-linguistic domain, pointing toward a domain-general conflict monitoring system (for an overview, see [3]). In the current study, it is investigated whether adaptations between syntactic and non-syntactic conflicts—as reported by Kan et al. [4]—replicate.[1]

Cognitive conflict has been identified as a key signal for the cognitive system that triggers processing adjustments between controlled and automated processing styles. Traditional views of information processing often contrast (i) controlled processing—which typically results in slow and resource-demanding information processing—and (ii) automated processing—resulting in fast and resource-saving information processing, but therefore being more error-prone in certain situations. However, there was no clear suggestion regarding how the brain knows when to implement which processing style. The conflict monitoring model [6] suggested that conflict monitoring and current conflict level are critical in determining the processing style adopted. According to this model, our cognitive system monitors information processing for the occurrence of conflict and constantly adjusts processing style to the current conflict level. For example, suppose we are driving on a motorway, and there is no decision to make regarding navigation. This might be a low-conflict situation that allows a somewhat automated processing style. However, if road signs indicate a required action (e.g. exit at the next junction), while a fellow passenger suggests an alternative course of action, such conflicting information might require a more controlled processing style. For example, we might reduce our speed to read the upcoming road signs or check our sat-nav more carefully.

In experimental psychology, such conflict situations and their consequences for subsequent behaviour are typically studied using paradigms such as the Stroop, Flanker or the Simon tasks or investigating errors as a conflict source (see [7–12]). Although these tasks are distinct, they share certain high-level features. For example, each involves two sources of information: specifically, information relevant to performing the task, and task-irrelevant information. When the task-relevant and task-irrelevant information both indicate the same required response, it is said to be compatible (low-conflict), whereas when the task-relevant and task-irrelevant information indicate different responses, it is said to be incompatible (high-conflict). For example, in a standard Stroop task, participants respond to the font colour (relevant information) of words (e.g. red, green, blue) with word meaning being irrelevant. In a standard Flanker task, participants respond to the central stimulus within a stimulus array (e.g. <<<<<, >><>>, HHHHH, SSHSS). Here, the central target (task-relevant) can match (compatible) or mismatch (incompatible) with the surrounding stimulus items. Similarly, in the Simon task, participants respond to a feature of the target stimulus (e.g. colour), that is presented on the left or right side of the screen. Trials are classified as compatible when the presentation side on the screen (left versus right) matches the required response hand (left versus right), and incompatible if the presentation side mismatches the response hand. Across these tasks, responses are typically slower and more error-prone in incompatible compared with compatible trials. However, what does such conflict trigger with regard to information processing adjustments in the subsequent trial? The core findings in this research area are summarized in the following paragraphs.

Interestingly, Gratton & colleagues [13] demonstrated that previous trial compatibility influences the current trial response. Gratton reported that conflict effects are reduced in the current trial if the previous

---

[1]I want to highlight that there has been another replication study also investigating whether the effects reported by Kan et al. [4] replicate. This study is already available, including results as a preprint, and failed to replicate the core effects (see [5]).

trial was a high-conflict trial. Botvinick *et al.* [6] further analysed and formalized these trial-to-trial adaptation patterns and developed the so-called conflict monitoring model. According to this model, response conflict is detected in incompatible trials, which results in a subsequent increase of cognitive control that eases the processing of further incompatible (i.e. high-conflict) trials, but slows the processing of compatible (i.e. low-conflict) trials. In the following years, these conflict adaptation patterns have been highly investigated, and also some alternative explanations have been brought forward (cf. [14]). For example, Mayr *et al.* highlighted that a specific confound of stimulus/response repetitions exists in many studies investigating conflict adaptation, and following the exclusion of these confounds, the conflict adaptation pattern disappears or is greatly reduced (for details, see [14–16]). However, there are also studies pointing to conflict adaptation patterns beyond sequential contingencies and stimulus repetitions (e.g. [17]).

Recently, conflict adaptation patterns have also been observed within the linguistic domain using negation as a source of conflict (e.g. 'not left' versus 'now left; 'not up' versus 'now up'), showing that conflict adaptation effects can be observed beyond the standard Stroop, Flanker and Simon task conflict domains [18,19]. Additionally, after extensive investigation of within-task conflict adaptation, research has recently extended to investigate whether conflict adaptation exists between different task types (e.g. [20]). This is especially interesting, first as between-task conflict adaptation patterns typically exclude the confound of direct stimulus repetitions, and second, such effects would point to conflict adaptation generalizing between different processing systems. Evidence in favour of between-task conflict adaptation transfer has been interpreted as evidence in favour of a domain-general conflict monitoring system. Today, the evidence in this research line is rather mixed, whereby some studies report transfer effects between certain task-types (for an overview, see [3]), whereas other studies do not report such effects and point toward rather specific conflict-control units [21–23].

Probably one of the transfer effects between the most diverse types of tasks reported in the literature was by Kan *et al.* [4]. They showed that transfer effects could be observed from a standard reading task—if the linguistic input triggered a re-analysis process via a syntactic ambiguity—and a standard Stroop task. Participants read syntactically ambiguous (e.g. *The basketball player accepted the contract would have to be negotiated*; incongruent) and non-ambiguous (e.g. *The basketball player accepted that the contract would have to be negotiated*; congruent) sentences [24] intermixed with performing a Stroop task. Previous fMRI studies showed that similar brain regions are involved during conflicting Stroop trials and comprehending syntactically ambiguous sentences [25,26], which led Kan *et al.* to the hypothesis that the cognitive control system might deal with both incongruencies in a domain-general approach. The tasks were pseudo-randomly intermixed during the experimental blocks. Kan *et al.* found that the conflict originating via sentence reading subsequently triggers conflict adaptation effects in the Stroop task. No transfer effects were reported from the Stroop conflict on the sentence reading task. Overall, this finding of transfer effects between such different tasks (reading garden path sentences and Stroop task) is in line with the suggestion by Braem *et al.* [3] that fully different tasks might result in transfer effects, as well as fully identical tasks, whereby medium tasks consisting of medium similarity result in no transfer effects.

This specific replication is relevant for several reasons. First, for understanding the architecture of cognition, it is important to know whether conflict monitoring works in a domain-general manner, as conflict and conflict adaptation effects are among the most studied phenomenon in experimental psychology and studies investigating cross-task transfer effects are still in a minority. Second, further evidence supporting a domain-general control system would indicate that similar principles work across different cognitive domains, supporting the neural reuse hypothesis of Anderson [27], whereby cognitive principles might work across various domains. Additionally, the results reported by Kan *et al.* were central to the development of the model of Braem *et al.* [3], and further studies are needed to corroborate this model regarding across-task transfer effects.

Here two replication studies are reported—first, the identical study reported by Kan *et al.* [4] in Experiment 1. Here, Kan *et al.* used a Stroop task that only involved stimulus conflict but no response conflict. Specifically, the task-irrelevant colours in the incompatible trials were not part of the response set (and therefore did not activate an alternative response). In a second experiment, the Stroop conflict was slightly different, specifically the distractor colours were part of the response set. This should result in larger conflict [28,29] and potentially larger cross-task transfer effects. Both experiments were pre-registered prior to data collection on aspredicted.org (https://aspredicted.org/blind.php?x=vt8mk9). This article received results-blind in-principle acceptance (IPA) at the Royal Society Open Science. Following IPA, the accepted Stage 1 version of the manuscript, not including results and discussion, was pre-registered on the OSF (https://osf.io/u726e). This pre-registration was performed after data analysis. All data and analysis scripts are available in OSF (https://osf.io/5btmg/).

# 2. Experiment 1

## 2.1. Method

### 2.1.1. Participants

To determine sample size, a power analysis based on the Kan *et al.* [4] study using the R package 'Superpower' (see [30]) was conducted. A simulation-based approach to determine the number of required participants with the desired power of 0.8 (0.9) resulted in 69 (91) for the previous by current interaction of interest. Specifically, the effect size was calculated from the partial eta squared value of the $2 \times 2$ interaction of previous sentence congruency and current Stroop congruency on reaction time (RT). One hundred participants were collected to achieve sufficient power of 0.9, given that some participants will need to be excluded (for details, see Kan *et al.* [4] and section *Data Analysis* below). All participants were recruited via the Amazon Mechanical Turk (MTurk) community, which is nowadays commonly used in behavioural research showing rather large compliance with the instructions [31]. Participants were paid the standard local rate for experiment participation (approx. 10 Euro per hour, converted to US dollars). Recruitment was limited to participants based in the United States, whose mother-tongue is English, which might be interesting as the original study was also US-based.[2] It was ensured that an individual participant can only take part in the experiment once. Additionally, participation in Experiment 1 excluded the opportunity to participate in Experiment 2 (and vice versa). Participants provided informed consent on the first page of the online study via putting a tick mark after reading the according section. The study was run under the ethics approval 2018_0831_132, approved by the 'Kommission für Ethik in der psychologischen Forschung' at the University of Tübingen.

### 2.1.2. Apparatus and stimuli

The experiment was written in JavaScript using JsPsych [32]. The experiment was hosted on a local server with the appropriate link being provided to participants via the MTurk website. Participants completed the experiment on their personal computer within the browser. Pilot-testing was performed using popular modern browsers (e.g. Chrome/Chromium, Firefox and Safari). Participants were requested not to use Internet Explorer, to use a standard desktop computer or laptop, and to complete the experiment in a quiet environment. Online testing of behavioural studies has, in the last years, gained significant importance, with various studies having shown that for experiments where factors are manipulated within-participants, online measurements can be treated as equivalent to lab-based studies (for the most recent test, see [33]). Also, simulation studies suggest that the additional technical noise from the online measurement of reaction times produces a negligible reduction of the effect size estimate and statistical power [34], suggesting that researchers' preconceptions regarding reaction time measurements in online experiments are misguided. In the present experiment, it was not possible to participate in the experiment via the use of a smartphone or a tablet-like device. An initial screen-size check at the start of the experiment ensures that each participant has a minimum screen resolution of approximately $1280 \times 720$ pixels. Stimulus size was controlled by a calibration routine performed at the start of the experiment. The calibration routine involved adjusting the size of a rectangular shape using the mouse until the size of the rectangle matched that of a regular bank card.

The stimulus materials are identical to those used by [4] within Experiment 1; specifically, the presentation of single-word Stroop stimuli (see [12]) and the word-by-word presentation (moving-window procedure) of whole sentences (see [35]). The Stroop stimuli consist of the words blue, green, yellow, brown, orange and red being presented in either blue, green, or yellow font colour. When font colour and word meaning match (e.g. the word 'blue' presented in blue), it is congruent, while when the font and the word meaning mismatch (e.g. the word 'blue' presented in brown), it is incongruent. Within this Stroop task, the task-irrelevant word meaning within incongruent trials does not belong to the response set (i.e. stimulus-based conflict). Within the sentence processing trials, participants press the spacebar to reveal one word at a time. Sentence materials are of two types: (i) experimental items and (ii) filler items. The experimental items are based on materials from Garnsey *et al.* [24] and consist of both temporarily unambiguous (congruent) or temporarily ambiguous (incongruent) items.[3]

---

[2]The other replication study by Aczel *et al.* [5] tested participants in Singapore, Australia and UK, and of course effects should typically persist across different native speakers.

[3]The stimulus material was kindly provided by the Kan and colleagues.

The Sentence task involved a moving-window procedure with materials adapted from Garnsey *et al.* [24]. Some items ($n = 21$) were difficult to process (ambiguous/incongruent) while others ($n = 21$) were easy to process (unambiguous/ congruent).

Sentence examples:

(a) The basketball player accepted that the contract would have to be negotiated. (unambiguous/congruent)
(b) The basketball player accepted the contract would have to be negotiated. (ambiguous/incongruent)

The sentences are presented using the moving-window procedure with the participant pressing the spacebar to reveal the next word.

## 2.1.3. Procedure

The experiment began with a screen informing participants that their browser would enter 'full-screen' mode, followed by the screen-size calibration routine described above. Participants are then required to give some basic demographic details (age, gender, handedness) and to click a check-box to indicate informed consent. It is not possible to continue with the experiment until this information is completed. The block/trial procedure followed that described by Kan *et al.* [4] and involved several phases. The first phase consisted of a training block with 18 Stroop trials in order to learn the response mapping. The mapping of keys (G, H, J) is randomly assigned to a response colour (blue, green, yellow)[4] for each individual participant, with participants instructed to respond with the ring, middle and index fingers of their dominant hand. The trial sequence involved the presentation of a fixation cross for 500 ms followed by the presentation of the Stroop stimulus. The Stroop stimulus remained on the screen until response (or 1000 ms elapsed). Feedback ('Correct', 'Error') was presented for 750 ms following the response, followed by a blank inter-trial interval (ITI) of 1000 ms. The training phase was followed by a baseline Stroop block consisting of 144 trials. No response feedback was provided during the baseline Stroop block. Following the baseline Stroop block, a single sentence material was presented. This item was randomly selected from the 11 available practice sentence items and was implemented to familiarize participants with the moving-window procedure. This procedure began with the presentation of the sentence mask (all letters replaced by the underscore character, figure 1). The participant was required to press the spacebar using their non-dominant hand to reveal the sentence word-by-word (see figure 1). Words that have already been revealed are subsequently re-masked. The next word appeared immediately following the spacebar response. There was no time limit to respond to the sentence items. This single trial sentence phase was followed by 20 randomly intermixed Stroop trials ($N = 10$) and sentence trials ($N = 10$). Again, participants responded to the Stroop items with their dominant hand and the sentence items with their non-dominant hand. The main experiment phase involved 197[5] intermixed Stroop and sentence trials. The sentence materials consisted of 21 congruent-type sentences, 21 incongruent-type sentences and 29 filler sentences.[6] Following 10 of the filler sentences, a True/False type comprehension question was presented. Participants responded using f and t keys on the keyboard with the response mapping being clearly indicated on the screen. All materials were pseudo-randomized presented with the single constraint that an experimental sentence was always followed by a Stroop task.[7]

---

[4]Please notice that in the original Kan *et al.* study, the keys were not randomly assigned between participants but kept in a fixed order.

[5]In the study of Kan *et al.* [4], they used 60 congruent and 60 incongruent Stroop trials, plus 21 congruent and 21 incongruent sentences which results in 162 trials. Adding the filler sentences this results in $162 + 29 = 191$ trials. In the present paper the test phase consisted of 63 congruent and 63 incongruent Stroop trials. This change was made to balance the possible combinations of relevant and irrelevant dimensions within the Stroop task. This is not something that seems to be balanced within the original study. For example, within subjects the word 'red' did not appear in blue the same number of times as did orange. Whilst this is not critical to the hypothesis under investigation, it is the reason for the slight change in the number of Stroop stimuli.

[6]Of the 42 congruent/incongruent-type sentences, there was both a congruent and an incongruent version of each sentence (see figure 1). Each participant only saw one version of each sentence—resulting in each participant seeing 21 congruent and 21 incongruent sentences—with congruent/incongruent sentence types split across two material sets that were counter-balanced across participants. This with all in line with the Kan *et al.* [4] study.

[7]The original study of Kan *et al.* used a single pseudo-randomized presentation order across all participants. This deviation in the current set-up was used as it is more standard in experimental research to newly randomize trial order for each participant (in order to exclude potential confounds of order).

The _____ _____ _____ _____ _____ _____ __ __ _____

___ basketball _____ _____ _____ _____ _____ __ __ _____

___ _____ player _____ ____ ___ _____ _____ ____ __ __ _____

**Figure 1.** Display example, each trial starts with an empty line indicating only word and sentence length. Subsequently with each button press the next word appears on the line.

### 2.1.4. Data analysis

The exclusion of participants from the described analysis is based on criteria set prior to data collection and includes several steps. First, the experiment phase involves the presentation of 29 filler items, 10 of which are followed by a comprehension question requiring a True/Yes or False/No response. Kan *et al.* [4] excluded participants ($n = 1$) with chance-level performance in the comprehension task. The remaining participants had an accuracy level above 70%. In the present study—in line with Kan *et al.*—therefore, participants who answered correctly on only six or fewer of these questions are excluded from the analysis, resulting in participants who remained in the analysis having above 70% accuracy rates.

The critical analyses involve the analysis of error rates and RTs of the Stroop trials in the experimental task. First, a direct follow-up of the original Kan *et al.* [4] analyses was conducted. Here, the critical analyses are the $2 \times 2$ repeated measure ANOVAs on the dependent variables RTs and error rates with the factors current Stroop level congruency (congruent/incongruent) and previous sentence congruency (congruent/incongruent). The expected pattern of results is in line with the conflict adaptation effect: specifically, reduced conflict following incongruent trials (i.e. smaller Stroop effect following ambiguous sentences).

*Reaction times*: Stroop RTs include correct trials only. For clarity, it needs to be mentioned that Kan *et al.* [4] conducted two ANOVAs for the RTs, one with outlier values replaced and one with outliers removed. They report in a footnote that both produced similar patterns and reported only the version using the replacement procedure within the main manuscript. The outlier replacement procedure reported by Kan *et al.* [4] involved replacing RTs 2.5 s.d.s beyond a participant's mean across all conditions with the 2.5 s.d. cut-off value. Therefore, the first ANOVA of the RTs follows exactly this procedure with using all RTs in the Stroop trials to get a good estimate of the deviation. The second ANOVA is identical but is performed using a trial exclusion procedure based on the 2.5 s.d.s criterion. The second ANOVA is conducted with the data excluding the outliers. Thus, both ANOVAs are expected to show similar patterns for a successful replication.

*Error rates*: For the error rate analysis, Kan *et al.* [4] again performed two analyses. One involved arcsin-transformed error rates and one with the raw proportions of errors. In their results section, the statistical analysis of the raw proportion was reported, but again it was mentioned earlier in the paper that a similar pattern was observed for the arcsin-transformed data. Here, both analyses are conducted and are expected to show a similar result pattern.

The Kan *et al.* [4] paper found domain-general adaptation across all dependent variables (RTs and error rates). Nevertheless, prior to data analysis, it was decided that if the current replication study shows discrepancies between the RT and error rate analysis, more weight would be given to the RT data regarding whether the results from the original study replicated or not. This procedure is in line with the tradition in the conflict adaptation literature where predominantly the RT data are reported (typically in line with the error rate data, but not always accompanied by the report of the error data; e.g. [21]). However, it should be acknowledged that such an approach might oversee potential meaningful theoretical combinations of the two measures [36,37].

*Sentence analysis*: To be consistent with the analyses of Kan *et al.* [4], a region-based analysis of the sentence materials is also reported with the aim to test for the congruency effect within the sentences. Again, this analysis mirrors that of Kan *et al.* in that separate *t*-tests are performed on the residual reading times within the critical regions for congruent and incongruent sentences, adjusted for word length and the number of words within a region. Raw reading times that were greater than 2.5 s.d.s beyond a specific participant's mean, irrespective of condition, were replaced with a value equal to the 2.5 s.d.s cut-off value. Following this, the predicted reading times were estimated from the regression equation for each sentence region (according to region length). Finally, residual reading

*The frustrated tourists understood* | *that* | *the message* | *would mean* | *they couldn't go. (Congruent)*
*The frustrated tourists understood* | | *the message* | *would mean* | *they couldn't go. (Incongruent)*

**Figure 2.** An example of the region-based analysis with red being the ambiguous area and green being the disambiguating region.

times were calculated by taking the difference between the predicted and actual reading times. Overall, the test of the sentence reading times serves as a control regarding whether participants slow down during reading incongruent sentences and did not serve to test the critical hypothesis. Here an approach as illustrated in figure 2 was applied.

*Follow-up analysis*: Following the analysis procedures described above that closely follow those described by Kan *et al.* [4], additional analyses beyond the original paper are reported to further validate any potential findings, both from a side of data quality and a theoretical perspective. Specifically, as this study was conducted online, data quality—beyond the basic sentence comprehension checks reported by Kan *et al.* [4]—should be ensured. Here, to a certain level, the performance within the baseline Stroop task offers the possibility to achieve this. Note that the following exclusion criteria are not based on trials critical to the experimental hypothesis but rather on baseline Stroop trials. Also, these are additional tests that were not conducted in the original study but should allow thorough conclusions especially with regard to a potential null effect (see below for details). Such exclusion criteria include (a) the removal of participants whose baseline Stroop performance in terms of accuracy is poor (greater than 30% error rate across conditions), with further exclusions based on the basic Stroop effect and the conflict adaptation effects within the baseline Stroop task;[8] (b) no Stroop effect; (c) non-conflict adaptation effect. With these additional exclusion criteria (b) and (c), the main hypothesis of interest is tested (i.e. Stroop performance following congruent/incongruent sentences) using a subset of participants who show both standard Stroop effects and conflict adaptation effects within the baseline Stroop trials. If such an analysis does not show cross-task conflict adaptation, this would be additional valuable information for the reader. A final exclusion criterion—and subsequent subset analysis—includes (d) the removal of participants who do not show a sentence conflict effect (incongruent greater than congruent). Thus, the final analysis is run with only a subset that shows the sentence conflict effect in RTs.[9] These analyses with these additional exclusion criteria are not based on the data from the critical comparisons. Nevertheless, such strict exclusions will ensure data quality across all crucial task aspects and allow additional insights regarding the main hypothesis while at the same time ensuring the presence of basic effects. Otherwise, conclusions regarding hypothesis rejection might not be based on firm data if basic effects cannot be observed.

## 2.2. Results

### 2.2.1. Data pre-processing

The experiments were analysed in line with the originally approved analysis steps with any unforeseen—minor—changes to those in the approved methods clearly stated (i.e. see footnote 10: the exclusion of extreme RTs before calculating the s.d.s for data exclusion). The online data collection procedure resulted in the collection of an additional six participants beyond the planned and pre-registered sample size of an initial 100. These additional datasets were not included in the analysis. Thus, the full sample consisted of 100 participants ($M_{age} = 40.86$, s.d.$_{age} = 10.89$, 49 female, 92 right-handed). The pre-registered participant exclusion based on filler question accuracy (minimum 70% correct), resulted in the removal of five participants. For both Stroop and sentence trials, the pre-registered outlier analysis procedure replaced RTs beyond 2.5 s.d.s of an individual participant's mean with the

---

[8]If the overall sample does not show the baseline Stroop effect, the sample size will be stepwise reduced until it contains a sample showing the baseline Stroop effect. It is not anticipated that this will happen, nevertheless it should be ensured that we are able to measure basic conflict and conflict adaptation effects in the current replication (identical procedure for (c) and (d)). As anticipated these issues mainly did not occur. Therefore, follow-up tests were not performed in a stepwise manner reducing the sample, but even in a stricter manner excluding all participants not showing Baseline Stroop effects, within-task CSE or sentence incongruency effects.

[9]Again, it is not anticipated that this exclusion procedure is needed, as it is expected that the overall sample shows this sentence congruency effect. Nevertheless, sentence reading time effects might be less strong than the Stroop effect. Resulting power issues will be considered.

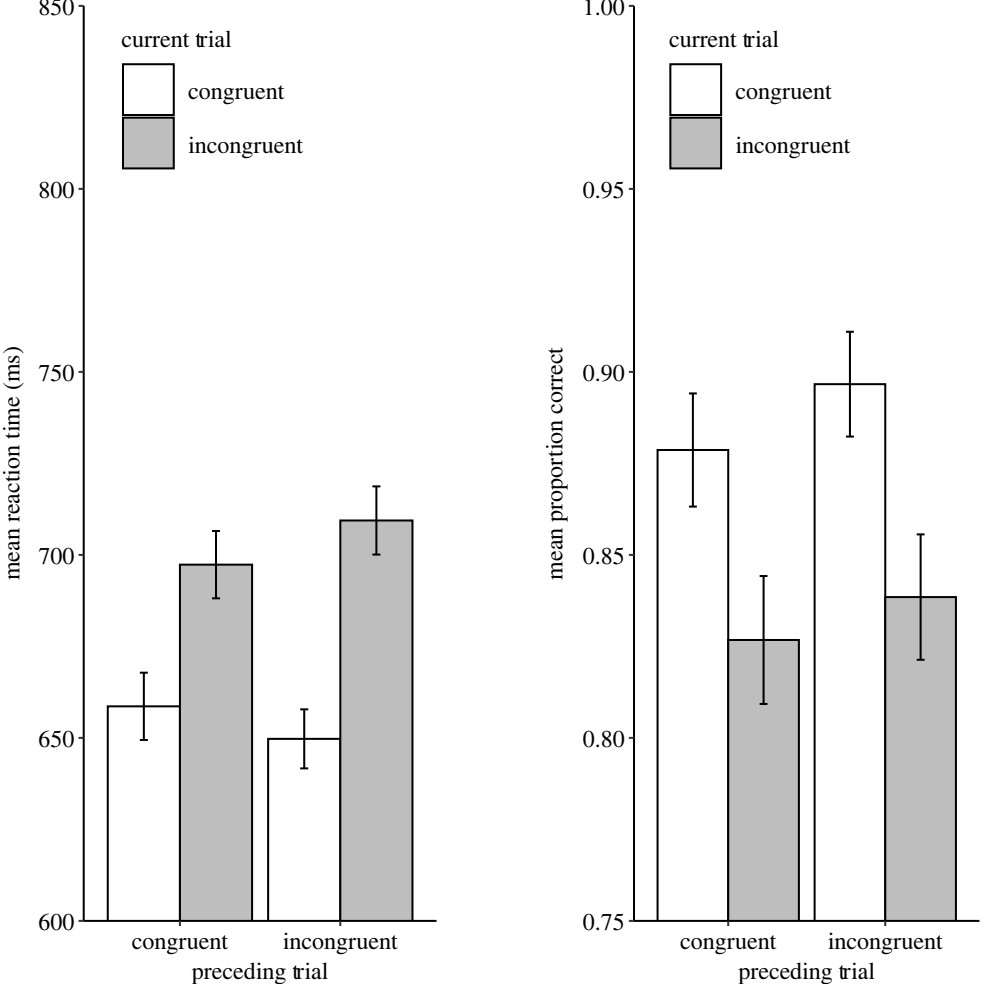

**Figure 3.** Mean RTs (left plot) and ERs (right plot) of Experiment 1. The error bars represent ± 1 s.e.m.

respective cut-off values.[10] As pre-registered, all RT analyses were performed on both the adjusted values and those based on an outlier removal procedure, and similarly, for error rate (ER) analyses, both arcsin and raw proportion values were analysed. For brevity, the equivalent statistics will only be reported for ANOVA results on the adjusted RT values, and the arcsin-transformed ERs[11] (all detailed analyses can be found in the analysis files). The identical analyses on unadjusted RTs (outliers removed) and analyses on raw proportion ERs are only reported if the result pattern differs in a theoretically relevant way. For ERs, raw proportions are reported in the figures and inline text.

### 2.2.2. Critical test (sentence to Stroop trials (RT and ER))

Two additional participants had to be removed from the subsequent analysis due to poor performance within the Stroop trials; specifically, they did not have any correct trials remaining in one (or more) of the analysis cells. Condition means for both RT and ER are displayed in figure 3 and summarized table 1 (appendix A). For RT, the ANOVA revealed a significant main effect of Stroop congruency with faster responses to congruent trials ($M = 654$ ms) compared with incongruent trials ($M = 703$ ms), $F_{1,92} = 87.37$, $p < 0.001$, $\eta_p^2 = 0.49$. The interaction between Previous Sentence Congruency and Current Stroop Congruency was significant, $F_{1,92} = 6.07$, $p = 0.016$, $\eta_p^2 = 0.06$. However, this interaction was driven by

---

[10]Visual inspection of the RT distribution revealed that a small proportion of trials had extremely long RTs (~0.17% > 2000 ms; $RT_{max} > 6$ min!). As inclusion of these values skews the calculation of the s.d., these values were excluded. It must be noted that this was not pre-registered, but the result pattern between the analyses with corrected values and the analyses that remove outlier trials was generally consistent.

[11]Please note that this deviated from the original pre-registration that was focused on the raw error rates. However, during the review process focus on the transformed error rates seemed of preference.

the congruency effect being larger following incongruent trials ($M = 60$ ms) than congruent trials ($M = 39$ ms), thus, showing a reversed congruency sequence effect (CSE). For ER, there was a significant main effect of Stroop Congruency with more correct responses to congruent trials ($M = 88.8\%$) compared with incongruent trials ($M = 83.3\%$), $F_{1,92} = 27.64$, $p < 0.001$, $\eta_p^2 = 0.23$. The interaction between Previous Sentence Congruency and Current Stroop Congruency was not significant, $F_{1,92} = 0.21$, $p = 0.647$. These critical tests do not support the main hypothesis regarding domain-general conflict adaptation.

### 2.2.3. Outcome neutral—sentence analysis

Following the procedure described by Kan *et al.* [4] and consistent with a sentence congruency effect, the critical regions showed a significant difference in both raw and residual reading times across the two critical regions (approx. 8–11 ms difference, $t$s(92) > 3.11, $p$s < 0.003, $d_z > 0.32$). Unexpectedly, there was a small (approx. 4 ms), but significant difference within the first region, $t$s(92) > 2.11, $p$s < 0.038. For full details, see table 2 (appendix A).

### 2.2.4. Follow-up analysis

*Baseline Stroop task (RT and ER)*: Five participants performed poorly within the Baseline Stroop task, committing more than 30% errors. This performance was not analysed in the original study. However, it is used here to further ensure participants' compliance with the task. Following the removal of these participants, the performance within the baseline Stroop task was analysed using a $2 \times 2$ repeated measures ANOVA with the within-participant factors Previous Congruency and Current Congruency. This analysis was performed to check within-task conflict adaptation patterns, again not analysed in the original paper. For RT, there was a main effect of Current Congruency with faster responses to congruent trials ($M = 563$ ms) compared with incongruent trials ($M = 609$ ms), $F_{1,89} = 251.97$, $p < 0.001$, $\eta_p^2 = 0.74$. The two-way interaction between Previous Congruency and Current Congruency was not significant, $F_{1,89} = 1.46$, $p = 0.231$, $\eta_p^2 = 0.02$, indicating that the congruency effect was not different following compatible trials ($M = 48$ ms) and incompatible trials ($M = 43$ ms). For ER, there was a main effect of Current Congruency with more accurate responses to congruent trials ($M = 89.6\%$) than to incongruent trials ($M = 84.7\%$), $F_{1,89} = 24.23$, $p < 0.001$, $\eta_p^2 = 0.21$. Similar to RT, the interaction between Previous Trial Congruency and Current Trial Congruency was not significant, $F_{1,89} = 0.06$, $p = 0.807$, $\eta_p^2 < 0.01$. Like RT, the congruency effect was similar following congruent ($M = 4.7\%$) and incongruent trials ($M = 5.0\%$), suggesting that the sample did not show within-task CSE effects.

*Subset analysis Sentence to Stroop trials (RT and ER)*: The critical tests reported above from Sentence to Stroop trials did not replicate the original result reported in Kan *et al.* [4]; specifically, there was no evidence—for RT nor ER—that the congruency effect in a Stroop trial was reduced following a difficult to process (incongruent) sentence type. However, several results must be highlighted. First, and somewhat surprisingly, the standard CSE was absent in both RT and ER within the Baseline Stroop block (note that this analysis was not reported in the original paper, therefore leaving open whether the within-task CSE was observed there). Secondly, although significant, the observed congruency effect within the Sentence trials was relatively small (approx. 10 ms) compared with the result reported within the original study (approx. 40 ms). Thus, further analyses were performed on subsets of the data according to additional performance criterion. The means are summarized in table 1 (appendix A). As highlighted within the Baseline Stroop analysis above, several participants demonstrated poor task performance. Removing these five participants from the Sentence to Stroop analysis again resulted in a significant interaction between Previous Sentence Type and Current Stroop Type for RT, $F_{1,88} = 4.03$, $p = 0.048$, $\eta_p^2 = 0.04$,[12] but not ER, $F_{1,88} = 0.10$, $p = 0.757$, $\eta_p^2 < 0.001$), with the mean patterns being similar to that reported for the full analyses (i.e. opposite pattern to the predicted CSE). Further removing both participants who demonstrated a lack of a basic Stroop effect ($N = 3$), and those participants showing no standard CSE in the Baseline Stroop task ($N = 46$),[13] resulted in a non-significant two-way interaction between Previous Sentence Congruency and Current Stroop Congruency for both RT, $F_{1,41} = 0.99$, $p = 0.327$, $\eta_p^2 = 0.02$ and ER, $F_{1,41} = 0.02$  0.01, $p = 0.900$, $\eta_p^2 < 0.01$). Finally, participants who did not

---

[12]The equivalent interaction using an outlier exclusion criterion was not significant, $F_{1,88} = 3.89$, $p = 0.052$, $\eta_p^2 = 0.04$.

[13]Although somewhat trivial, the analysis of the Baseline Stroop trials within this subset sample was significant, $F_{1,41} = 92.96$, $p < 0.001$, $\eta_p^2 = 0.69$, revealing a CSE of 42 ms.

demonstrate a congruency effect within the critical sentence regions were excluded ($N = 31$).[14] The resulting analyses revealed non-significant interactions for both RT, $F_{1,31} = 0.46$, $p = 0.503$, $\eta_p^2 = 0.01$) and ER, $F_{1,31} = 0.08$, $p = 0.777$, $\eta_p^2 < 0.01$.

## 2.3. Discussion

Experiment 1 aimed to replicate the first experiment by Kan *et al.* [4]. Kan *et al.* demonstrated that the Stroop congruency effect (faster RTs and fewer errors for congruent compared with incongruent trials) was reduced following difficult-to-process sentences. Specifically, they reported a standard CSE and argued that conflict experienced within the sentence reading task recruits control processes that reduce the influence of subsequent conflict—even across different task domains—thus providing evidence for domain-general cognitive control. The results of the current experiment do not provide additional support for this finding. Indeed, the critical two-way interaction between Previous Sentence Congruency and Current Stroop Congruency was significant in the opposite direction in the RT data. However, one might argue that the current results are difficult to interpret for two reasons. First, the full sample did not exhibit a standard within-task CSE.[15] Second, the observed congruency effect within the Sentence task was smaller than that observed within the original study. As a counter-argument to such observations, follow-up analyses 'cherry-picking' subsets of participants showing both a standard within-task Stroop CSE and larger Sentence congruency effects also did not provide evidence for cross-task CSEs. However, note that these procedures resulted in a decrease in sample size and, therefore, a loss in power.

# 3. Experiment 2

Experiment 2 served as a conceptual replication of Experiment 1 of Kan *et al.* [4] with one specific change regarding the Stroop stimulus material: instead of using a purely stimulus-based Stroop conflict, the Stroop conflict originated from additional response conflict. Specifically, the irrelevant word meaning in an incongruent Stroop trial was now part of the response set, which should result in larger Stroop conflict as both a semantic and a response competition mechanism contribute to the size of the Stroop effect (e.g. [38,39]). Given that this manipulation should result in larger Stroop effects, the opportunity to observe a reduced-Stroop effect following ambiguous sentences should also be enlarged.

## 3.1. Methods

### 3.1.1. Participants

The power analysis as described in Experiment 1 was used, resulting in a planned sample size of 100 participants. Participant recruitment was as described for Experiment 1.

### 3.1.2. Apparatus/design

Identical to Experiment 1, with the only change being that for the incongruent Stroop trials, colours from the response set were used (see Procedure).

### 3.1.3. Procedure

Experiment 2 was identical to Experiment 1 except for the following change: the Stroop task was changed to a response-based conflict task. Specifically, the Stroop task involved the presentation of the words blue, green or yellow in blue, green or yellow font colour. Thus, the irrelevant word meaning was part of the response set. This should result in larger conflict in the Stroop task [28,29]. Again, the practice Stroop block consisted of 12 trials, while the baseline Stroop block consisted of 144 trials. The experimental phase consisted of 120 Stroop trials.

---

[14]Again somewhat trivially, the analysis of the critical word regions within this subset sample was significant, $ts(31) > 4.65$, $ps < 0.001$, with a congruency effect of ~16 to 20 ms.

[15]It must be noted that the original study did not actually report within-task CSE for the baseline Stroop task, leaving open whether it was observed there.

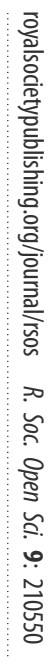

**Figure 4.** Mean RTs (left plot) and ERs (right plot) of Experiment 2. The error bars represent ± 1 s.e.m.

## 3.2. Results

### 3.2.1. Data pre-processing

The online data collection procedure resulted in the collection of an additional 13 participants. Like in Experiment 1, these additional datasets were disregarded. Thus, the full sample consisted as planned of 100 participants ($M_{age}$ = 40.66, s.d.$_{.age}$ = 10.96, 50 female, 93 right-handed). The pre-registered participant exclusion based on filler question accuracy (minimum 70% correct), resulted in the removal of four participants. All other data-preprocessing procedures were as described in Experiment 1.[16]

### 3.2.2. Critical test (sentence to Stroop trials (RT and ER))

One additional participant was removed from the subsequent analysis due to poor performance within the Stroop trials; specifically, they did not have any correct trials remaining in one of the analysis cells. Condition means for both RT and ER are displayed in figure 4. For RT, the ANOVA revealed a significant main effect of Stroop Congruency with faster responses to congruent trials ($M$ = 669 ms) compared with incongruent trials ($M$ = 748 ms), $F_{1,94}$ = 211.94, $p < 0.001$, $\eta_p^2$ = 0.69. The interaction between Previous Sentence Congruency and Current Trial Stroop Congruency, although not significant, $F_{1,94}$ = 3.12, $p = 0.081$, $\eta_p^2$ = 0.03, showed numerically a larger congruency effect following congruent trials ($M$ = 86 ms) than incongruent trials ($M$ = 73 ms). For ER, there was a significant main effect of Stroop congruency with more correct responses to congruent ($M$ = 87.3%) than incongruent trials ($M$ = 76.8%), $F_{1,94}$ = 57.41, $p < 0.001$, $\eta_p^2$ = 0.38. Like RT, there was no evidence that the congruency effect was

---

[16]Visual inspection of the RT distribution again revealed that a small proportion of trials had extremely long RTs (~0.18% > 2000 ms; $RT_{max}$ > 45 s!), which were subsequently excluded from the s.d. calculations to avoid bias.

modulated by Previous Sentence Congruency, $F_{1,94} = 1.69$, $p = 0.197$, $\eta_p^2 < 0.002$. Together, these tests do not support the idea of cognitive control adjustments acting in a domain-overarching manner.

### 3.2.3. Outcome neutral—sentence analysis

Again, following the procedure described by Kan *et al.* [4] and adopted within Experiment 1, the critical regions within the sentences showed a significant difference in both raw and residual reading times (approx. 11 ms difference) within the first critical region, $ts(94) > 4.03$, $ps < 0.001$, $d_z > 0.41$. Unlike Experiment 1, the difference within the second critical region (approx. 4 ms) was not significant, $ts(94) < 1.43$, $ps > 0.157$. The unexpected difference observed within the pre-critical region within Experiment 1 was not replicated here (approx. 1 ms, $ts(94) < 1$). For full details, see table 2 (appendix A).

### 3.2.4. Follow-up analysis

The rationale for the subsequently reported follow-up analyses is consistent with that reported within Experiment 1. Specifically, as planned in the pre-registration—as follow-up analyses—stricter participant exclusion criteria were applied according to Baseline Stroop performance (ER and presence of an RT CSE) and the sentence congruency effect (presence of a congruency effect within the critical regions).

*Baseline Stroop task (RT and ER)*: Five participants performed poorly within the Baseline Stroop phase of the experiment, committing more than 30% errors. Following the removal of these participants, the Baseline Stroop performance was analysed using a 2 × 2 repeated measures ANOVA with the within-participant factors Previous Congruency and Current Congruency. For RT, there was a main effect of Current Congruency with faster responses to congruent trials ($M = 575$ ms) compared with incongruent trials ($M = 627$ ms), $F_{1,90} = 153.58$, $p < 0.001$, $\eta_p^2 = 0.63$. The two-way interaction between Previous Congruency and Current Congruency was significant, $F_{1,90} = 28.58$, $p < 0.001$, $\eta_p^2 = 0.24$, indicating a larger congruency effect following congruent trials (66 ms) than incongruent trials (39 ms), suggesting that this time within-task CSE effects are present. For ER, there was a main effect of Current Congruency with more accurate responses to congruent trials (89.1%) than to incongruent trials (76.2%), $F_{1,90} = 96.17$, $p < 0.001$, $\eta_p^2 = 0.52$. Similar to RT, the interaction between Previous Trial Congruency and Current Trial Congruency was also significant, $F_{1,90} = 9.99$, $p = 0.002$, $\eta_p^2 = 0.10$, with the congruency effect being larger following congruent trials ($M = 16.9\%$) than incongruent trials ($M = 9.0\%$).

*Subset analysis Sentence to Stroop trials (RT and ER)*: The main analysis from Sentence to Stroop trials did not support the original result reported in Kan *et al.* [4]; specifically, there was no evidence—for RT or ER—that the congruency effect was reduced following a difficult-to-process (incongruent) sentence type. However, it can be noted that numerically—at least for RT—the size of the congruency effect was smaller following incongruent compared with congruent trials (CSE: 13 ms). Unlike Experiment 1, the standard CSE effect was observed—both for RT and ER—within the Baseline Stroop block. Similarly to Experiment 1, the observed congruency effect within the sentence trials was again small numerically (approx. 11 ms in the first critical region) compared with the result reported within the original study (approx. 40 ms) (also in terms of effect size), but still significant. Further analyses were performed on subsets of the data according to additional performance criteria previously outlined. As highlighted within the Baseline Stroop analysis above, several participants demonstrated poor task performance. Removing these five participants from the Sentence to Stroop analysis again resulted in a non-significant interaction between Previous Sentence Type and Current Stroop Type for RT, $F_{1,90} = 3.64$, $p = 0.060$, $\eta_p^2 = 0.04$, and for ER, $F_{1,90} = 2.32$, $p = 0.131$, $\eta_p^2 = 0.03$), with the mean patterns being similar to that reported for the full analyses. Further removing both participants who demonstrated a lack of a basic Stroop effect ($N = 5$), and those participants showing no standard within-task CSE ($N = 22$),[17] again resulted in a non-significant two-way interaction between Previous Sentence Congruency and Current Stroop Congruency for RT, $F_{1,65} = 3.56$, $p = 0.064$, $\eta_p^2 = 0.05$, but a significant interaction in ER, $F_{1,65} = 4.61$, $p = 0.035$, $\eta_p^2 = 0.07$. Finally, participants who did not demonstrate a congruency effect within the critical sentence regions were additionally excluded ($N = 33$).[18] The resulting analyses revealed a non-significant interactions for RT, $F_{1,44} = 1.17$, $p = 0.286$, $\eta_p^2 = 0.03$, but a significant interaction for ER, $F_{1,44} = 5.87$, $p = 0.020$, $\eta_p^2 = 0.12$. Regarding the interaction for ERs, the congruency effect was larger following congruent trials ($M = 11.9\%$) compared with incongruent trials ($M = 6.9\%$).

---

[17]Although somewhat trivial, the analysis of the Baseline Stroop trials within this subset sample was significant, $F_{1,65} = 93.31$, $p < 0.001$, $\eta_p^2 = 0.59$, revealing a CSE of 46 ms.

[18]Again, somewhat trivially, the analysis of the critical word regions within this subset sample was significant, $ts(44) > 4.80$, $ps < 0.001$, with a congruency effect of approximately 14 to 20 ms.

## 3.3. Discussion

Experiment 2—like Experiment 1—aimed to replicate the first experiment reported in Kan *et al*. [4]. Experiment 2 followed the same rationale as Experiment 1, with the only change being that the Stroop trials within Experiment 2 contained response conflict instead of stimulus conflict; specifically, in Experiment 2, the irrelevant word meaning was part of the response set. The core findings of Experiment 2 offer little to support the result reported by Kan *et al*. [4]. Although numerically there was evidence for a reduced congruency effect following incongruent sentence trials, this effect was small and not statistically significant. Additionally, this result was not mirrored in the error rate data of the main analysis. Like in Experiment 1, the sentence congruency effect within the critical regions was significant. Unlike Experiment 1, clear CSEs were demonstrated within the Baseline Stroop block, mitigating a potential argument that it is somewhat unlikely to observe cross-task adaptation with a participant sample who do not show within-task adaptation effects. Only in the follow-up analysis using reduced sample sizes based on baseline Stroop CSE and size of sentence incongruency effect was there a CSE effect but only within the ERs.

## 3.4. Experiment 1 and 2: combined analysis

As Experiments 1 and 2 are very similar in structure, a combined analysis was possible, and increases the power of the analysis, especially when considering the follow-up subset analyses where participant exclusions are high. Also, the true effect size might be smaller than the effect size indicated from the results of the original Kan *et al*. paper, which often seems to be case (see [40]). The subsequent analyses followed those detailed in Experiments 1 and 2, and provide the main interpretation in the case of previously diverging results.

### 3.4.1. Critical test (sentence to Stroop trials (RT and ER))

For RT, the ANOVA revealed a significant main effect of Stroop congruency with faster responses to congruent trials (662 ms) compared with incongruent trials (726 ms), $F_{1,187} = 267.87$, $p < 0.001$, $\eta_p^2 = 0.59$. The interaction between Previous Sentence Congruency and Current Stroop Congruency was not significant, $F_{1,187} = 0.53$, $p = 0.467$, $\eta_p^2 < 0.01$. For ER, there was a significant main effect of Stroop congruency with more correct responses to congruent (88.0%) than incongruent trials (80.0%), $F_{1,187} = 82.11$, $p < 0.001$, $\eta_p^2 = 0.31$. Like RT, there was no evidence that the Stroop congruency effect was modulated by Previous Sentence Congruency, $F_{1,187} = 0.25$, $p = 0.616$, $\eta_p^2 < 0.01$. Thus, this combined analysis does not support the idea of domain-general conflict adaptation.

### 3.4.2. Outcome neutral—sentence analysis

The critical regions within the sentences showed a significant difference in both raw and residual reading times within both the first critical region (approx. 11 ms, $ts(187) > 5.95$, $ps < 0.001$, $d_z = 0.434$) and the second critical region (approx. 6 ms, $ts(187) > 3.03$, $ps < 0.003$). There was no significant difference within either the pre-critical region or the post-critical region ($ts(187) < 1.07$, $ps > 0.290$).

### 3.4.3. Follow-up analysis

*Baseline Stroop task (RT and ER)*: Ten participants performed poorly within the baseline Stroop phase of the experiment, committing more than 30% errors. Following the removal of these participants, the baseline Stroop performance was analysed using a $2 \times 2$ repeated measures ANOVA with the within-participant factors Previous Congruency and Current Congruency. For RT, there was a main effect of Current Congruency with faster responses to congruent trials (569 ms) compared with incongruent trials (618 ms), $F_{1,180} = 364.28$, $p < 0.001$, $\eta_p^2 = 0.67$. The two-way interaction between Previous Congruency and Current Congruency was significant, $F_{1,194} = 22.03$, $p < 0.001$, $\eta_p^2 = 0.11$, indicating a larger congruency effect following congruent trials ($M = 57$ ms) than incongruent trials ($M = 41$ ms) in line with the idea of within-task CSE. For ER, there was a main effect of Current Congruency with more accurate responses to congruent trials ($M = 89.3\%$) than to incongruent trials ($M = 80.4\%$), $F_{1,180} = 103.76$, $p < 0.001$, $\eta_p^2 = 0.37$. Unlike RT, the interaction between Previous Trial Congruency and Current Trial Congruency was not significant, $F_{1,180} = 3.59$, $p = 0.060$, $\eta_p^2 = 0.02$. However, numerically, the congruency effect was larger following congruent trials ($M = 10.9\%$) than incongruent trials ($M = 7.0\%$).

*Subset analysis Sentence to Stroop trials (RT and ER)*: The critical test from Sentence to Stroop trials did not support the original result reported in Kan *et al*. [4], with non-significant interactions between Previous Sentence Type and Current Stroop Congruency for both RT and ER. In this combined analysis, the standard CSE effect was observed within the baseline Stroop block. However, the observed congruency effect within the sentence trials was again small (approx. 11 ms), but still significant. Further analyses were

performed on subsets of the data according to the same additional performance criterion used within Experiment 1 and 2. As highlighted within the Baseline Stroop analysis above, several participants demonstrated poor task performance. Removing these 10 participants from the Sentence to Stroop analysis again resulted in a non-significant interaction between Previous Sentence Type and Current Stroop Type for RT, $F_{1,179} = 0.06$, $p = 0.811$, $\eta_p^2 < 0.01$, and for ER, $F_{1,179} = 0.59$, $p = 0.442$, $\eta_p^2 < 0.001$. Further analyses removing both participants who demonstrated a lack of a basic Stroop effect ($N = 8$), and those participants showing no standard within-task CSE ($N = 68$), again resulted in a non-significant two-way interaction between Previous Sentence Congruency and Current Stroop Congruency for RT, $F_{1,107} = 0.40$, $p = 0.530$, $\eta_p^2 < 0.01$, and ER, $F_{1,107} = 2.13$, $p = 0.147$, $\eta_p^2 = 0.02$. Finally, participants who did not demonstrate a congruency effect within the critical sentence regions were additionally excluded ($N = 64$). The resulting analyses revealed non-significant interactions for RT, $F_{1,76} = 0.13$, $p = 0.717$, $\eta_p^2 < 0.01$, and for ER, $F_{1,76} = 2.36$, $p = 0.129$, $\eta_p^2 = 0.03$.

# 4. General discussion

The present study aimed to replicate Experiment 1 of Kan *et al.* [4] which demonstrated that reading incongruent sentences (i.e. garden-path sentences) leads to the recruitment of control processes. Specifically, the Stroop effect was reduced following incongruent compared with congruent sentences, in line with the idea that conflict adjustments operate in a domain-overarching manner. In Experiment 1 of the present study, a direct replication of the original Kan *et al.* Experiment 1 was attempted, while in Experiment 2, a conceptual replication was attempted using a Stroop task with response-based conflict rather than stimulus-based conflict. The study was implemented according to the pre-registered plan and was successful in terms of overall data quality. Participants followed task instructions (high accuracy within the filler comprehension questions), demonstrated relatively clear congruency effects within the garden-path sentences, and the Stroop data was as expected (slower RTs and increased ERs in incongruent compared with congruent trials). To further ensure data quality regarding the critical hypothesis (i.e. a two-way interaction between Previous Sentence Congruency and Current Stroop Congruency), several additional tests of this interaction were made using specific subsets of the original sample. In these analyses, the participants were 'cherry-picked' according to performance in the baseline Stroop task (i.e. presence of a Stroop effect and standard CSE), and sentence processing conflict (i.e. did the participants show slower reading times within incongruent compared with congruent sentences within the critical regions). Across all these analyses, no converging evidence in favour of the cross-domain CSE could be found in the RT data and the ER data, except for an effect in the ER data for the subset analyses within Experiment 2 only. It must be noted that the ER effect was not evident in the combined analysis of Experiments 1 and 2. Taken together, the present study did not replicate the cross-domain CSE effect reported in Experiment 1 of Kan *et al.* [4] (see also previous replication by Aczel *et al.* [5]).

What limitations could mitigate the present findings? First, the data collection for the present study was conducted online. Although online data collection has been treated with scepticism in the past from both reviewers and editors due to increased technical and situational variance [41,42], recently—and accelerated by the COVID19 pandemic—continued research has provided converging evidence across various experimental paradigms, but also improved technical infrastructure (e.g. recruitment via MTurk, hosting platforms, experimental implementation), that web-based data collection cannot be viewed as inferior. This especially applies to within-participant designs, which are, in large aspects, highly comparable to laboratory-based studies [43–46]. Especially relevant for the current paper are findings showing that research in the area of the Stroop task allows conclusive findings comparable to lab studies [47–49], even if data collection is administered via MTurk [50]. Nevertheless, it cannot be taken for granted that web-based studies can replace laboratory-based studies across all domains. To ensure a certain level of data quality in the present study, additional analyses aimed to show that certain basic effects were present in the data (some beyond the effects analysed in the original study, for example, the baseline within-task CSE effect in the baseline Stroop task) that should allow additional conclusions regarding the validity of the replication attempt. Crucial in this aspect was the presence of the incongruency effect in the sentence reading task, which was present across all analysis steps and both experiments. Also, in the follow-up analysis, limiting the included participants (i.e. participants showing within-task CSE effects and the sentence incongruency effect) still did not provide any converging evidence for cross-domain CSEs. Of course, one can argue that within the subset analyses, statistical power is reduced. However, even the most restrictive participant subset in the combined analysis of the experiments still would have reached above 0.80 power ($N = 77$).

Thus, from the overall data patterns in the present experiments, there is no further evidence in favour of domain-general cognitive control adjustments. Perhaps this is a somewhat surprising result, as one can argue that both tasks contain certain verbal aspects and therefore are not even the most stringent testing ground for

investigating cross-task conflict adjustments. Indeed, the original study by Kan *et al*. [4] found conflict adjustments in even more diverse tasks (Experiment 3 using Nekercube reversals, but see also the failed replication attempt of [5]). Interestingly, Braem *et al*. [3] suggest—based on a literature review—that medium task similarity does not result in transfer effects between tasks in terms of cross-task CSEs, and one could argue that the present study falls into such a medium similarity category. However, crucially in the Braem *et al*. [3] paper, the Kan *et al*. study was categorized as evidence for large context dissimilarity resulting in cross-task transfer effects, and therefore forms a crucial data point for the proposed U-shaped relationship between task-similarity and transfer effects. Therefore, this U-shaped relationship between task-similarity and the to-be-observed transfer effects might need to be re-evaluated following converging evidence against the findings in the Kan *et al*. [4] paper (see present replication attempt, but also [5,51]).

There is another interesting aspect that is worth discussing against the backdrop of the original Kan *et al*. [4] study and the follow-up studies in the recent years investigating domain-general cognitive control adjustments by investigating sentence processing (e.g. [52,53]). This research—in line with the present study—is often relevant not only for questions regarding domain-general control adjustments but also for questions regarding control adjustments within language processing itself. Surprisingly, it is still rather unclear whether control adjustments can be observed within a pure language comprehension paradigm—for example, if incongruent sentences are followed by other congruent or incongruent sentences. In other words: if we shift towards a more linguistic reading task (e.g. reading incongruent sentences, whereby the incongruency could be triggered by semantic or syntactic ambiguities, but also basic negation) can we still observe CSE-like patterns between sentences? It seems that this is a fundamental question that also needs further research. Indeed, recent literature seems somehow mixed regarding this question. For example, reading semantically ambiguous sentences did not affect reading times of subsequent ambiguous sentences [51]. However, the N400 language marker was influenced by both global and local manipulations of congruency when reading world-knowledge and semantic violations [54]. One observation that may explain the above discrepancy is the length of the sentence materials. In the study of Simi *et al*. [51], the items consist of longer, more complex sentences, whereas in the study using the N400 marker, every item is a short, simple three-word sentence (e.g. *Ladybirds are stripy*). Thus, the time interval between where conflict is experienced in trial $N - 1$, and the subsequent conflict in trial N may play a role [55]. This observation was also highlighted within the original Kan study where they analysed Stroop to sentence sequences and did not observe any CSEs. Thus, future research might also need to focus on questions regarding within-task CSE effects during language processing.

Another aspect to consider regarding the original Kan *et al*. [4] study—and therefore the two replication studies—is the relatively small number of trials. Comparing these studies to the standard literature of within-task CSEs, there is a rather large discrepancy. In within-task CSE studies, often a very limited number of stimuli (and thus many stimulus-response repetitions) are used (for discussion on the often-ignored role of trial numbers, see [56]). Even in studies that aim to eliminate direct stimulus-response repetitions, the number of trials is still relatively high compared with the present study (e.g. 800 trials in [17]). Another issue that needs to be mentioned is the potential over-estimation of effect size in the original study (for discussion, see [57]). If this is indeed the case, this would suggest that the power calculations that served as the basis for the replication attempts resulted in too small a sample size. However, given that the combined analysis—highly increasing sample size—also did not show any evidence in favour of a cross-domain CSE, it seems unlikely that pure power problems lie at the core of the failed replication.

Was there any evidence that can be taken in favour of a domain-general conflict adaptation mechanism in the present study? If one assumes that the original finding by Kan *et al*. [4] was not a chance finding, it would be interesting to see what analysis in the present or previous replication attempts support the original result. In the present analyses, there was a subset analysis within Experiment 2—selecting the participants showing good baseline Stroop performance and the presence of a within-task CSE—that showed a significant CSE in the analysis of the ER. Given that the previous replication found a significant effect on the ER in the main analysis [5]—and interpreted this as potential evidence for cross-domain CSE—it seems worth mentioning this finding here. However, although this exclusion procedure was based on alternative performance metrics in the present experiment, such an approach—especially when considered alongside the RT data from the same analyses—offers little evidence supporting domain generality. Also, the combined analysis did not show such an effect in ERs—even following the identical participant exclusion procedures—again speaking against the relevance of this finding for the present conclusions.

# 5. Conclusion

What overall conclusions can we draw from the present study? In summary, there was no evidence supporting the domain generality of conflict adjustments when investigating the interplay between reading garden-path

sentences and the Stroop task using either stimulus (Experiment 1) or response-based (Experiment 2) conflict. In comparison with the previous replication [5], which showed partial support (in ER data only) for domain generality of conflict adjustments in this set-up, the present replication was Internet-based, an issue that can always open doors for specific criticism (but see discussion above). On the other side, Internet-based data collection allowed the participant sample to be based in the USA—in line with the original study—which might have been a plus point given that the original participant sample was USA-based. There is still the possibility that the demographics of the present sample differed in a crucial way from the original sample. However, there is currently—to my knowledge—no theoretical basis suggesting that domain generality of conflict adjustments should be limited to specific participants samples. In the past years, there has been another attempt to conceptually replicate the Kan *et al.* [4] study using a semantic conflict [51]. This study also could not replicate the original findings despite increasing the number of trials and having clear effects within baseline measurements. Thus, as it currently stands, the domain generality in terms of between task adjustments from reading to Stroop tasks should probably be interpreted with care.

Ethics. The study was run under the ethics approval 2018_0831_132, approved by the 'Kommission für Ethik in der psychologischen Forschung' at the University of Tübingen.

Data accessibility. All data and analysis scripts are available on OSF (https://osf.io/5btmg/).

Authors' contributions. C.D.: formal analysis, funding acquisition, investigation, methodology, project administration, resources, software, visualization, writing—original draft, writing—review and editing.

Conflict of interest declaration. We declare we have no competing interests.

Funding. This work was funded by the Deutsche Forschungsgemeinschaft (DFG, German Research Foundation)—specifically a Heisenberg Fellowship appointed to Carolin Dudschig (419439493 / 419433647).

# Appendix A

See tables 1 and 2.

**Table 1.** Mean RTs and ERs on the Stroop trials following congruent or incongruent sentences. *Note.* Exp = Experiment, Exp 1 = Experiment 1, Exp 2 = Experiment 2, Comb = combined experiments. CC = congruent sentence followed by congruent Stroop, CI = congruent sentence followed by incongruent Stroop, IC = incongruent sentence followed by congruent Stroop, II = incongruent sentence followed by incongruent Stroop.

| Exp | analysis | condition | RT | | accuracy | |
| --- | --- | --- | --- | --- | --- | --- |
| | | | *M* | s.d. | *M* | s.d. |
| Exp1 | full (*N* = 93) | CC | 659 | 88 | 87.9 | 14.9 |
| | | CI | 697 | 87 | 82.7 | 16.9 |
| | | IC | 650 | 77 | 89.7 | 13.8 |
| | | II | 709 | 88 | 83.8 | 16.5 |
| | | CSE | −21 | | −0.7 | |
| | subset a (*N* = 89) | CC | 659 | 87 | 89.6 | 11.0 |
| | | CI | 698 | 88 | 84.1 | 14.8 |
| | | IC | 650 | 78 | 91.1 | 11.6 |
| | | II | 706 | 88 | 85.6 | 13.7 |
| | | CSE | −17 | | 0.0 | |
| | subset b/c (*N* = 42) | CC | 653 | 91 | 89.1 | 11.8 |
| | | CI | 694 | 97 | 83.8 | 14.6 |
| | | IC | 646 | 89 | 92.6 | 9.1 |
| | | II | 700 | 89 | 87.4 | 11.9 |
| | | CSE | −13 | | 0.1 | |
| | subset d (*N* = 32) | CC | 647 | 95 | 89.7 | 10.9 |
| | | CI | 691 | 106 | 85.3 | 14.6 |
| | | IC | 639 | 92 | 93.0 | 7.9 |
| | | II | 692 | 92 | 89.6 | 11.7 |
| | | CSE | −9 | | 1.0 | |

(Continued.)

| Exp | analysis | condition | RT | | accuracy | |
|---|---|---|---|---|---|---|
| | | | *M* | s.d. | *M* | s.d. |
| Exp 2 | full (*N* = 95) | CC | 664 | 72 | 88.4 | 14.0 |
| | | CI | 750 | 89 | 76.9 | 18.6 |
| | | IC | 674 | 86 | 86.2 | 15.1 |
| | | II | 747 | 89 | 76.7 | 19.9 |
| | | CSE | 13 | | 2.0 | |
| | subset a (*N* = 91) | CC | 664 | 73 | 89.3 | 13.6 |
| | | CI | 751 | 89 | 77.6 | 18.5 |
| | | IC | 673 | 87 | 86.8 | 14.4 |
| | | II | 746 | 90 | 77.6 | 19.7 |
| | | CSE | 9 | | 2.5 | |
| | subset b/c (*N* = 66) | CC | 655 | 64 | 90.2 | 12.4 |
| | | CI | 744 | 90 | 77.9 | 16.1 |
| | | IC | 668 | 85 | 87.9 | 14.1 |
| | | II | 739 | 86 | 79.2 | 17.8 |
| | | CSE | 18 | | 3.6 | |
| | subset d (*N* = 45) | CC | 650 | 68 | 92.0 | 11.2 |
| | | CI | 738 | 95 | 80.1 | 17.3 |
| | | IC | 657 | 89 | 89.7 | 12.6 |
| | | II | 733 | 88 | 82.8 | 16.9 |
| | | CSE | 12 | | 5.0 | |
| Comb | full (*N* = 188) | CC | 661 | 80 | 88.2 | 14.4 |
| | | CI | 724 | 92 | 79.8 | 18.0 |
| | | IC | 662 | 82 | 87.9 | 14.5 |
| | | II | 728 | 90 | 80.2 | 18.6 |
| | | CSE | −3 | | 0.7 | |
| | subset a (*N* = 180) | CC | 661 | 80 | 89.4 | 12.3 |
| | | CI | 725 | 92 | 80.8 | 17.1 |
| | | IC | 661 | 83 | 88.9 | 13.2 |
| | | II | 726 | 91 | 81.5 | 17.4 |
| | | CSE | −2 | | 1.2 | |
| | subset b/c (*N* = 108) | CC | 655 | 75 | 89.7 | 12.1 |
| | | CI | 725 | 96 | 80.2 | 15.7 |
| | | IC | 659 | 87 | 89.8 | 12.6 |
| | | II | 724 | 89 | 82.4 | 16.2 |
| | | CSE | 4 | | 2.1 | |
| | subset d (*N* = 77) | CC | 649 | 80 | 91.0 | 11.1 |
| | | CI | 719 | 102 | 82.3 | 16.4 |
| | | IC | 650 | 90 | 91.1 | 11.0 |
| | | II | 716 | 92 | 85.6 | 15.3 |
| | | CSE | 2 | | 3.2 | |

**Table 2.** Mean reading times in congruent and incongruent sentences. Note. Exp = Experiment, Exp 1 = Experiment 1, Exp 2 = Experiment 2, Comb = combined experiments.

| Exp | analysis | sentence | word position | | | | | | | | | |
|---|---|---|---|---|---|---|---|---|---|---|---|---|
| | | | ...accepted | | that | | the contract | | would have | | to be | |
| | | | M | s.d. | M | s.d. | M | s.d. | M | s.d. | M | s.d. |
| Exp 1 | full (N = 93) | congruent | 329 | 103 | 328 | 101 | 313 | 97 | 319 | 100 | 365 | 121 |
| | | incongruent | 333 | 107 | — | — | 324 | 100 | 328 | 107 | 364 | 122 |
| | subset d (N = 32) | congruent | 344 | 113 | 338 | 105 | 320 | 102 | 330 | 106 | 373 | 131 |
| | | incongruent | 351 | 118 | — | — | 340 | 105 | 347 | 115 | 378 | 129 |
| Exp 2 | full (N = 95) | congruent | 331 | 115 | 327 | 105 | 311 | 100 | 321 | 101 | 374 | 124 |
| | | incongruent | 330 | 117 | — | — | 322 | 101 | 325 | 108 | 375 | 125 |
| | subset d (N = 45) | congruent | 328 | 103 | 326 | 96 | 305 | 90 | 315 | 91 | 377 | 128 |
| | | incongruent | 330 | 103 | — | — | 325 | 93 | 331 | 104 | 382 | 130 |
| Comb | full (N = 188) | congruent | 330 | 109 | 328 | 103 | 312 | 98 | 320 | 100 | 370 | 122 |
| | | incongruent | 332 | 112 | — | — | 323 | 100 | 327 | 107 | 369 | 123 |
| | subset d (N = 77) | congruent | 334 | 107 | 331 | 100 | 311 | 95 | 321 | 97 | 376 | 129 |
| | | incongruent | 338 | 109 | — | — | 331 | 98 | 338 | 108 | 380 | 129 |

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
