## [Peer Review File · Royal Society Open Science]

Review History

RSOS-201814.R0 (Original submission)

Review form: Reviewer 1 (Charlotte Eben)

Do you have any ethical concerns with this paper?

No

Have you any concerns about statistical analyses in this paper?

No

Recommendation?

Accept with minor revision

Comments to the Author(s)

The author proposes a study to directly and conceptually replicate the Experiment 1 by Kan et al. (2013). In my opinion this is a sound replication proposal and I am impressed by the details the author provides especially with regard to the online testing procedure.

I have only a few minor suggestions you might want to consider:

- In my opinion there are details about the statistical model, the factors and levels and the inference criteria missing in the data analysis plan but also in the pre-registration.
- For me the theoretical motivation for the conceptual replication (Experiment 2) is not entirely clear.
- Has the congruent/incongruent procedure been tested online? I am a huge fan of online testing but from my experience I would advise to run a short pilot to see if you can find the effect in the sentence reading task.
- The original study does not have a big sample size, it might be possible that the original effect size is overestimated due to a lack of power in the original study. Maybe this is something to keep in the back of your mind?
- In my experience you might encounter some technical difficulties while data collection as well. What do you plan on doing with data sets where single trials are missing (e.g. due to instable internet connection)? What will you do with participants that start the experiment again because they might have accidentally closed the window?
- Maybe consider using Prolific instead of Mturk? Mturk is not made specifically for academic research whereas Prolific is (Palan & Schitter, 2018; <https://doi.org/10.1016/j.jbef.2017.12.004>). Of course, this is not at all required, I just have very good experience with Prolific and the way they try to ensure data quality.
- De Leuw (2015) himself suggests using Chrome as this is best tested in jsPsych, it also seems that Chrome (in combination with Windows) “adds the least random noise to RT measurement” (see Pronk et al. 2020; <https://doi.org/10.3758/s13428-019-01321-2>).
- The combination of data sets as described in the pre-registration might be interesting here as well.
- In the spirit of open science and to make it easier to follow your analyses pipeline you could also share the code and the analyses scripts together with the pre-registration or at the later stage together with the data.

And some very minor points:

- On p. 7 in line 20/21 it has to be Braem instead of Bream.
- On p. 10 line 22: do you mean “right” finger or ring finger? Because for left-handed people it is the left finger?

Review form: Reviewer 2

Do you have any ethical concerns with this paper?

No

Have you any concerns about statistical analyses in this paper?

No

Recommendation?

Accept with minor revision

Comments to the Author(s)

Summary

The manuscript is a stage 1 replication study of the paper:

Kan, I. P., Teubner-Rhodes, S., Drummey, A. B., Nutile, L., Krupa, L., & Novick, J. M. (2013). To adapt or not to adapt: The question of domain-general cognitive control. *Cognition*, 129(3), 637-651.

The author proposes to replicate Experiment 1 from Kan et al., as well as conducting a conceptual replication using a small modification. Experiment 1 of Kan et al. had participants perform intermixed trials of a sentence comprehension and Stroop task. The observation of a two-way interaction between preceding (sentence) congruency and Stroop task congruency was interpreted as evidence for domain general conflict adaptation – the Stroop effect was smaller when it followed an ambiguous sentence. The original Experiment 1 used Stroop stimuli where the incongruent words ('brown', 'orange', 'red') did not match the font colours to be named ('blue', 'green', 'yellow'). The conceptual replication proposed here (Experiment 2) modifies this so that they do overlap.

Review

I found the proposal to be generally clear and well-written. I thought that the primary and secondary criteria were met, or nearly met with a potential small elaboration (point 1). The proposed replication appears to match the details of the original experiment, with the perhaps notable exception that it will be conducted with an online sample. I was satisfied with the author's justification for why this does not compromise the ability of the proposed study to detect the effect of interest. The author proposes numerous secondary analyses which I thought were sufficient quality control checks. I have a couple of minor comments on the logic of the conceptual replication study, but they are probably not critical.

1) I thought that there could be a more explicit statement about the critical hypothesis test(s), as it may not be obvious to a reader not familiar with Kan et al. From the power analysis, I assume that the main test is the two-way interaction between previous-trial and current-trial congruency. Some elaboration may be warranted on whether there is an expected direction/pattern of condition means. Further, the author proposes to analyse RTs, error rates, and arcsin transformed error rates (as in the original). With multiple outcomes, there is potential flexibility in determining whether the original findings replicated. It might be worth commenting on whether significant effects in one or all outcomes will be considered as sufficient evidence.

2) The conceptual replication seems sensible on the surface, though it did raise a couple of questions in my mind that the author might want to consider.

The reasoning given is that Experiment 2 "...should result in larger conflict... and potentially larger cross-task transfer effects". One interpretation of this is that conflict needs to be sufficiently large for transfer to occur. Another is that transfer could occur in both experiments, but the effect size would be larger in Experiment 2. If it is the latter, what is the benefit of the additional Experiment over increasing the power of Experiment 1 for a smaller effect size? I don't object to the planned sample size, and it is already substantially larger than the original study. However, my reading is that it is based on the original effect size, and prior replication efforts have indicated that it is common for replications to produce smaller effect sizes than the original studies (e.g. Open Science Collaboration, 2015).

3) Kan et al. make the distinction between representational conflict and response conflict (Egner, 2008), and focus on representational conflict throughout their experiments. In their Stroop tasks, and in Experiment 1 here, the incongruent words were not part of the eligible response set. Experiment 2 here aims to elicit a larger conflict effect by adding response conflict (the incongruent words are part of the eligible response set). The author may want to comment on

whether response conflict should also be expected to contribute to the transfer effect. Naively, I have no reason to expect that it should not, but the prediction doesn't necessarily follow from the original study.

4) Just to bring it to the author's attention if they are not already aware - there is preprint available online of another proposal to replicate Kan et al.: <https://psyarxiv.com/5k8rq/>

References

Egner, T. (2008). Multiple conflict-driven control mechanisms in the human brain. *Trends in Cognitive Sciences*, 12, 374–380.

Open Science Collaboration. (2015). Estimating the reproducibility of psychological science. *Science*, 349, aac4716. doi:<https://doi.org/10.1126/science.aac4716>

Review form: Reviewer 3 (Balazs Aczel)

Do you have any ethical concerns with this paper?

No

Have you any concerns about statistical analyses in this paper?

Yes

Recommendation?

Reject

Comments to the Author(s)

Before going into details of the review, I have to mention that I am in a special situation as I have been requested to be a reviewer of this submission while I have a Stage 2 submission under review at RSOS. In fact, my team has replicated the very same study, Kan et al., 2013.

An advantage of the registered report format that we can give advice to the submitters before too much investment in the project has been done. Saying that I would discourage the author to conduct the replication, or in the proposed form for the following reasons:

We replicated Exp 1 from Kan et al. in three countries by independent labs with greater sample size. Compared to the proposed work, we very closely followed the original protocol and conducted the experiments in the lab and not online (some other deviations of the proposal are discussed below).

Our results were rather discouraging as you can read it in our preprint: https://psyarxiv.com/5k8rq

Besides all of this, even the original study of Kan et al. (2013) raised doubts that Exp. 1 is a good test of the theory, that is why they (and we) conducted Exp. 2 and 3. but the submission doesn't propose to replicate them. Nevertheless, our results were similarly pessimistic about the presence of congruency sequence effect, making us believe that the cross-task congruency sequence effect is either nonexistent or the design of Kan et al. is not a good test of the effect.

In short, I would encourage the author to use a different design to test to effect. My lab is open for further discussion about the potential empirical directions.

Below, I list a few more observations about the proposed design.

Is the proposed Exp1 a direct replication?

I think there are a few deviations from the original study. I do not mind them saying that it is a direct replication but it should be apparent that there are deviations.

Online vs in lab

“The mapping of keys (G, H, J) is randomly assigned to a response colour (blue, green, yellow) for each individual participant...” The original study doesn’t randomize it.

Page 10 line 59 (footnote): I believe this is an important deviation from the original study, however, I am not sure I understand how the stimulus set looked like. There were 42 non-filler sentences in the original study. The footnote suggests that participants did not see all the 42 sentences but just one pseudo-randomized set of 21 sentences. In the original study, the participants saw all the 42 sentences. As the materials for Experiment 1 of Kan et al. 2013 are available the specific order of the sentences in the original study could be applied in the replication as well.

Page 11 line 26: I am not sure that I understand the reasoning behind the exclusion criteria on the comprehension probe task. The author mentions Kan excluding only one participant at a chance level (50%) and that all the other participants score above 70%. The author then states that the replication will follow Kan et al. (2013) and participants with 6 or fewer correct responses will be dropped. This indicates a 60% (given that there were 10 comprehension probe sentences) cutoff level which is neither 50% nor 70%.

The whole sentence preprocessing part is missing. This is not a problem in itself but should be noted as a deviation and justified.

Page 10 line 52: The test part consists of 197 trials. In Kan they used 60 cong stroop + 60 incong stroop + 21 cong and 21 incong sentences which is 162. If the filler sentences are included that is $162 + 29 = 191$.

Page 11 line 59 (footnote): The stepwise reduction and testing for the presence of the Stroop effect are problematic if NHST is used to test the presence of the effect. Continuous testing can modify the alpha level hence the results.

General comments

Page 10 line 55: The started sentence is not finished.

Page 11 line 15: The footnote mark could be moved to the sentence regarding the RT outlier exclusions.

Page 7 line 32: I am not sure that I am convinced that Exp 2 is needed based on this one-sentence explanation.

Page 7 line 55: What was the exact effect size used for the sample size determination?

There is nothing about the analysis procedure in the paper. Nor about the inferences that the authors will make based on the results. Again, a lot of questions remain unanswered.

Decision letter (RSOS-201814.R0)

Dear Dr Dudschig,

The Editors assigned to your Stage 1 Replication submission ("Are control processes domain-general? A replication of "To adapt or not to adapt? The question of domain-general cognitive control" (Kan et al., 2013)") have now received comments from reviewers. We would like you to revise your paper in accordance with the referee and editors suggestions which can be found below (not including confidential reports to the Editor). Please note this decision does not guarantee eventual acceptance.

Please submit a copy of your revised paper within three weeks (i.e. by the 01-Jan-2021). If deemed necessary by the Editors, your manuscript will be sent back to one or more of the original reviewers for assessment. If the original reviewers are not available we may invite new reviewers.

When submitting your revised manuscript, you must respond to the comments made by the referees and upload a file "Response to Referees" in the "File Upload" step. Please use this to document how you have responded to the comments, and the adjustments you have made. In order to expedite the processing of the revised manuscript, please be as specific as possible in your response.

Once again, thank you for submitting your manuscript to Royal Society Open Science and we look forward to receiving your revision. If you have any questions at all, please do not hesitate to get in touch. Full author guidelines may be found at <https://royalsocietypublishing.org/rsos/replication-studies#AuthorsGuidance>.

on behalf of Professor Chris Chambers (Registered Reports Editor, Royal Society Open Science)
openscience@royalsociety.org

Associate Editor Comments to Author (Professor Chris Chambers):

Three specialist reviewers have now assessed the Stage 1 manuscript. Reviewers 1 and 2 are broadly positive, judging that the Stage 1 primary criteria are largely met and recommend in-principle acceptance (IPA) following minor revision. Reviewer 3, however, judges that neither criteria are met and recommends rejection and a re-evaluation of the project.

This was an unusual review process because, by coincidence, RSOS is currently assessing a Replication submission of the same original study, co-authored by Reviewer 3 (signed by Balazs Aczel). The Aczel paper is currently under Stage 2 review following the completion of the work, and so I felt it would be sensible to invite the main author of that replication to assess your replication proposal. The reviewer has kindly made their Stage 2 manuscript available as a preprint (which was also flagged by Reviewer 2), and as you will see from his assessment, based upon on the results of their (larger) replication attempt, the reviewer believes it is futile to re-attempt it here.

From an editorial standpoint, whether the prior replication attempt succeeded or failed (or even whether it happened at all) is not relevant to the accept/reject decision regarding the current Stage 1 submission. The submissions are in that sense independent. However, I feel that Reviewer 3's assessment is particularly valuable in this case because knowledge of the prior replication outcomes may lead you to reassess whether your proposed replication is the best use of your resources. In addition, Reviewer 3 is also concerned about deviations from the original study methodology (primary criterion #1), and the validity/robustness of the proposed replication attempt (primary criterion #2). As the reviewer notes, many key details are missing from the manuscript and need to be clarified.

Provided all concerns that related to the primary criteria are addressed then the current Stage 1 submission can achieve IPA following revision (again, regardless of the prior replication study). However, in light of the outcomes of this prior replication, you may decide to take Reviewer 3's advice and re-evaluate the current approach, perhaps even working with Reviewer 3 (which is possible -- in the event that authors end up collaborating with a reviewer as part of a Registered Report or Replication submission then that reviewer is simply removed from the review process moving forward).

I appreciate that this is an unusual and somewhat complicated peer review process and remain open to informal discussion with the authors concerning specific scenarios moving forward (feel free to email me directly at chambersc1@cardiff.ac.uk or via the RSOS email address).

Reviewer Comments to Author:

Reviewer: 1

Comments to the Author(s)

The author proposes a study to directly and conceptually replicate the Experiment 1 by Kan et al. (2013). In my opinion this is a sound replication proposal and I am impressed by the details the author provides especially with regard to the online testing procedure.

I have only a few minor suggestions you might want to consider:

- In my opinion there are details about the statistical model, the factors and levels and the inference criteria missing in the data analysis plan but also in the pre-registration.
- For me the theoretical motivation for the conceptual replication (Experiment 2) is not entirely clear.
- Has the congruent/incongruent procedure been tested online? I am a huge fan of online testing but from my experience I would advise to run a short pilot to see if you can find the effect in the sentence reading task.
- The original study does not have a big sample size, it might be possible that the original effect size is overestimated due to a lack of power in the original study. Maybe this is something to keep in the back of your mind?
- In my experience you might encounter some technical difficulties while data collection as well. What do you plan on doing with data sets where single trials are missing (e.g. due to instable internet connection)? What will you do with participants that start the experiment again because they might have accidentally closed the window?
- Maybe consider using Prolific instead of Mturk? Mturk is not made specifically for academic research whereas Prolific is (Palan & Schitter, 2018; <https://doi.org/10.1016/j.jbef.2017.12.004>). Of course, this is not at all required, I just have very good experience with Prolific and the way they try to ensure data quality.
- De Leuw (2015) himself suggests using Chrome as this is best tested in jsPsych, it also seems that Chrome (in combination with Windows) "adds the least random noise to RT measurement" (see Pronk et al. 2020; <https://doi.org/10.3758/s13428-019-01321-2>).
- The combination of data sets as described in the pre-registration might be interesting here as well.

- In the spirit of open science and to make it easier to follow your analyses pipeline you could also share the code and the analyses scripts together with the pre-registration or at the later stage together with the data.

And some very minor points:

- On p. 7 in line 20/21 it has to be Braem instead of Bream.

- On p. 10 line 22: do you mean “right” finger or ring finger? Because for left-handed people it is the left finger?

Reviewer: 2

Comments to the Author(s)

Summary

The manuscript is a stage 1 replication study of the paper:

Kan, I. P., Teubner-Rhodes, S., Drummey, A. B., Nutile, L., Krupa, L., & Novick, J. M. (2013). To adapt or not to adapt: The question of domain-general cognitive control. *Cognition*, 129(3), 637-651.

The author proposes to replicate Experiment 1 from Kan et al., as well as conducting a conceptual replication using a small modification. Experiment 1 of Kan et al. had participants perform intermixed trials of a sentence comprehension and Stroop task. The observation of a two-way interaction between preceding (sentence) congruency and Stroop task congruency was interpreted as evidence for domain general conflict adaptation – the Stroop effect was smaller when it followed an ambiguous sentence. The original Experiment 1 used Stroop stimuli where the incongruent words (‘brown’, ‘orange’, ‘red’) did not match the font colours to be named (‘blue’, ‘green’, ‘yellow’). The conceptual replication proposed here (Experiment 2) modifies this so that they do overlap.

Review

I found the proposal to be generally clear and well-written. I thought that the primary and secondary criteria were met, or nearly met with a potential small elaboration (point 1). The proposed replication appears to match the details of the original experiment, with the perhaps notable exception that it will be conducted with an online sample. I was satisfied with the author’s justification for why this does not compromise the ability of the proposed study to detect the effect of interest. The author proposes numerous secondary analyses which I thought were sufficient quality control checks. I have a couple of minor comments on the logic of the conceptual replication study, but they are probably not critical.

1) I thought that there could be a more explicit statement about the critical hypothesis test(s), as it may not be obvious to a reader not familiar with Kan et al. From the power analysis, I assume that the main test is the two-way interaction between previous-trial and current-trial congruency. Some elaboration may be warranted on whether there is an expected direction/pattern of condition means. Further, the author proposes to analyse RTs, error rates, and arcsin transformed error rates (as in the original). With multiple outcomes, there is potential flexibility in determining whether the original findings replicated. It might be worth commenting on whether significant effects in one or all outcomes will be considered as sufficient evidence.

2) The conceptual replication seems sensible on the surface, though it did raise a couple of questions in my mind that the author might want to consider.

The reasoning given is that Experiment 2 “...should result in larger conflict... and potentially larger cross-task transfer effects”. One interpretation of this is that conflict needs to be sufficiently large for transfer to occur. Another is that transfer could occur in both experiments, but the effect

size would be larger in Experiment 2. If it is the latter, what is the benefit of the additional Experiment over increasing the power of Experiment 1 for a smaller effect size? I don't object to the planned sample size, and it is already substantially larger than the original study. However, my reading is that it is based on the original effect size, and prior replication efforts have indicated that it is common for replications to produce smaller effect sizes than the original studies (e.g. Open Science Collaboration, 2015).

3) Kan et al. make the distinction between representational conflict and response conflict (Egner, 2008), and focus on representational conflict throughout their experiments. In their Stroop tasks, and in Experiment 1 here, the incongruent words were not part of the eligible response set. Experiment 2 here aims to elicit a larger conflict effect by adding response conflict (the incongruent words are part of the eligible response set). The author may want to comment on whether response conflict should also be expected to contribute to the transfer effect. Naively, I have no reason to expect that it should not, but the prediction doesn't necessarily follow from the original study.

4) Just to bring it to the author's attention if they are not already aware - there is preprint available online of another proposal to replicate Kan et al.: <https://psyarxiv.com/5k8rq/>

References

- Egner, T. (2008). Multiple conflict-driven control mechanisms in the human brain. *Trends in Cognitive Sciences*, 12, 374–380.
- Open Science Collaboration. (2015). Estimating the reproducibility of psychological science. *Science*, 349, aac4716. doi:<https://doi.org/10.1126/science.aac4716>

Reviewer: 3

Comments to the Author(s)

Before going into details of the review, I have to mention that I am in a special situation as I have been requested to be a reviewer of this submission while I have a Stage 2 submission under review at RSOS. In fact, my team has replicated the very same study, Kan et al., 2013.

An advantage of the registered report format that we can give advice to the submitters before too much investment in the project has been done. Saying that I would discourage the author to conduct the replication, or in the proposed form for the following reasons:

We replicated Exp 1 from Kan et al. in three countries by independent labs with greater sample size. Compared to the proposed work, we very closely followed the original protocol and conducted the experiments in the lab and not online (some other deviations of the proposal are discussed below).

Our results were rather discouraging as you can read it in our preprint:
<https://psyarxiv.com/5k8rq>

Besides all of this, even the original study of Kan et al. (2013) raised doubts that Exp. 1 is a good test of the theory, that is why they (and we) conducted Exp. 2 and 3. but the submission doesn't propose to replicate them. Nevertheless, our results were similarly pessimistic about the presence of congruency sequence effect, making us believe that the cross-task congruency sequence effect is either nonexistent or the design of Kan et al. is not a good test of the effect.

In short, I would encourage the author to use a different design to test to effect. My lab is open for further discussion about the potential empirical directions.

Below, I list a few more observations about the proposed design.

Is the proposed Exp1 a direct replication?

I think there are a few deviations from the original study. I do not mind them saying that it is a direct replication but it should be apparent that there are deviations.

Online vs in lab

“The mapping of keys (G, H, J) is randomly assigned to a response colour (blue, green, yellow) for each individual participant...” The original study doesn’t randomize it.

Page 10 line 59 (footnote): I believe this is an important deviation from the original study, however, I am not sure I understand how the stimulus set looked like. There were 42 non-filler sentences in the original study. The footnote suggests that participants did not see all the 42 sentences but just one pseudo-randomized set of 21 sentences. In the original study, the participants saw all the 42 sentences. As the materials for Experiment 1 of Kan et al. 2013 are available the specific order of the sentences in the original study could be applied in the replication as well.

Page 11 line 26: I am not sure that I understand the reasoning behind the exclusion criteria on the comprehension probe task. The author mentions Kan excluding only one participant at a chance level (50%) and that all the other participants score above 70%. The author then states that the replication will follow Kan et al. (2013) and participants with 6 or fewer correct responses will be dropped. This indicates a 60% (given that there were 10 comprehension probe sentences) cutoff level which is neither 50% nor 70%.

The whole sentence preprocessing part is missing. This is not a problem in itself but should be noted as a deviation and justified.

Page 10 line 52: The test part consists of 197 trials. In Kan they used 60 cong stroop + 60 incong stroop + 21 cong and 21 incong sentences which is 162. If the filler sentences are included that is $162 + 29 = 191$.

Page 11 line 59 (footnote): The stepwise reduction and testing for the presence of the Stroop effect are problematic if NHST is used to test the presence of the effect. Continuous testing can modify the alpha level hence the results.

General comments

Page 10 line 55: The started sentence is not finished.

Page 11 line 15: The footnote mark could be moved to the sentence regarding the RT outlier exclusions.

Page 7 line 32: I am not sure that I am convinced that Exp 2 is needed based on this one-sentence explanation.

Page 7 line 55: What was the exact effect size used for the sample size determination?

There is nothing about the analysis procedure in the paper. Nor about the inferences that the authors will make based on the results. Again, a lot of questions remain unanswered.

Author's Response to Decision Letter for (RSOS-201814.R0)

See Appendix A.

RSOS-201814.R1 (Revision)

Review form: Reviewer 1 (Charlotte Eben)

Do you have any ethical concerns with this paper?

No

Have you any concerns about statistical analyses in this paper?

No

Recommendation?

Accept in principle

Comments to the Author(s)

Thank you very much. All my comments have been addressed sufficiently.

Review form: Reviewer 2

Do you have any ethical concerns with this paper?

No

Have you any concerns about statistical analyses in this paper?

No

Recommendation?

Accept in principle

Comments to the Author(s)

The revised manuscript addresses my previous comments. I have one note on a rationale the author gives in their response, though this is not used in the manuscript so it may not warrant further changes.

My previous comment 1 suggested that the author could consider how they would interpret potentially different effects in reaction times and error rates. The manuscript now states that they focus on reaction times, as they are more commonly used in the literature. However their response to my comment also notes:

“Typically, in human information processing models the measures of reaction times and error rates are often seen as measuring the same underlying cognitive process (e.g. Draheim, Hicks, & Engle, 2016).”

I'm not sure this reference supports focusing on reaction times. Draheim et al. propose using a composite measure of RT and accuracy, on the basis that the latency costs typically used in task-switching are contaminated by speed-accuracy trade-offs. The way in which multiple processes contribute to patterns of RT and error effects is also something I have been interested in in my own work using evidence accumulation models (Hedge et al., 2018a, 2018b). For example, in models like the drift-diffusion model, an increase in the boundary separation parameter can lead to an increase in reaction time effects and a decrease (or no observable change) in error rates. This parameter has previously been implicated in conflict adaptation in the form of post-error slowing

(Dutilh et al., 2011). Though sequential congruency effects do not follow the same pattern, the point is that basing an interpretation on either reaction times or error rates may overlook theoretically meaningful combinations of the two.

References

Dutilh, G., Vandekerckhove, J., Forstmann, B. U., Keuleers, E., Brysbaert, M., & Wagenmakers, E. J. (2012). Testing theories of post-error slowing. *Attention, Perception, & Psychophysics*, 74(2), 454-465. <https://doi.org/10.3758/s13414-011-0243-2>

Hedge, C., Powell, G., Bompas, A., Vivian-Griffiths, S., & Sumner P. (2018a). Low and variable correlation between reaction time costs and accuracy costs explained by accumulation models: meta-analysis and simulations. *Psychological Bulletin*, 144(11), 1200-1227. <http://dx.doi.org/10.1037/bul0000164>

Hedge, C., Powell, G., & Sumner, P. (2018b). The mapping between transformed reaction time costs and models of processing in aging and cognition. *Psychology and Aging*, 33(7), 1093-1104. <http://dx.doi.org/10.1037/pag0000298>

Review form: Reviewer 3 (Balazs Aczel)

Do you have any ethical concerns with this paper?

No

Have you any concerns about statistical analyses in this paper?

Yes

Recommendation?

Accept with minor revision

Comments to the Author(s)

In this second round of reviews, I found a lot of improvement in the submission, especially in the analysis plan section. My big picture view is that I still think that if our own study did not replicate the evidence that the original paper showed for Exp1 then a new study would need more and not fewer participants to further explore this question. Since the present proposal is not planning to use Bayesian analysis, the expected non-significant results won't allow for any assessment of the original hypothesis. As a registration, I still miss the explicit links between result patterns and interpretations. Below, I explain these and a few other comments.

In their response, the author stated that "the data has already been collected". I might have got misled on that by the manuscript future tense wording, e.g., "Therefore 100 participants will be collected to achieve sufficient power of .9, given that some participants will need to be excluded.." I find it still confusing.

From our 152 participants, we lost 20 due to exclusions. Online data-collection might be more noisy. I would expect even more exclusions.

Now that the author knows that with a greater sample we could not fully replicate the effect, I wonder if the power analysis would need to be recalculated with our outputs.

I appreciate that an analysis part is added to this registered report. I think that is a crucial part. We can see that the author plans to conduct the main analyses with and without exclusions. It would be important to add how the author would interpret if the results are not in line, which analysis would be the basis of the conclusions.

I cannot seem to find how the sentences will be divided into sentence regions. That was a tough task for us. It would be important to have it worked out before the analysis.

The region-based analysis works with reading time predictions based on the person's reading times. It is crucial to state whether the prediction will be calculated with the inclusion of all the readings, or just the congruent sentences or the filler sentences as well.

I cannot see what pattern of results of the sentence region-based analysis will lead to what conclusions. Again, something crucial for a registration.

It would be great to add 'outcome-neutral analyses', for example to see if there is Stroop effect at all. Without that, the rest of the analyses are not too interesting.

Two experiments described but we don't know how the author would interpret the results if they contradict each other. What pattern of results would replicate the original findings and what would contradict the original hypothesis?

"was presented for xxx ms" is a misleading wording for online experiments. The truth would be something like "We used the xxx method for requesting that the computer present the stimulus for xxx ms" For an explanation, see <https://twitter.com/ceptional/status/1296686833305153536>

Balazs Aczel

Decision letter (RSOS-201814.R1)

Dear Dr Dudschig

On behalf of the Editors, I am pleased to inform you that your Manuscript RSOS-201814.R1 entitled "Are control processes domain-general? A replication of "To adapt or not to adapt? The question of domain-general cognitive control" (Kan et al., 2013)" deemed suitable for in-principle acceptance in Royal Society Open Science subject to minor revision in accordance with the referee and editor suggestions. Please find their comments at the end of this email.

The reviewers and handling editors have recommended publication, but also suggest some minor revisions to your manuscript. Therefore, I invite you to respond to the comments and revise your manuscript.

Please you submit the revised version of your manuscript within 7 days (i.e. by the 23-Feb-2021). If you do not think you will be able to meet this date please let me know immediately.

To revise your manuscript, log into <https://mc.manuscriptcentral.com/rsos> and enter your Author Centre, where you will find your manuscript title listed under "Manuscripts with

Decisions". Under "Actions," click on "Create a Revision." You will be unable to make your revisions on the originally submitted version of the manuscript. Instead, revise your manuscript and upload a new version through your Author Centre.

Full author guidelines can be found here <https://royalsocietypublishing.org/rsos/replication-studies#AuthorsGuidance>

on behalf of Professor Chris Chambers
(Subject Editor, Royal Society Open Science)
openscience@royalsociety.org

Associate Editor Comments to Author (Professor Chris Chambers):

The Stage 1 manuscript was returned to three reviewers who assessed the original submission. The reviews are broadly positive and we are now closer to IPA. Reviewer 1 is now satisfied but there remain some final issues to address concerning the consideration of RTs and error rates (Reviewer 2), and clarity and precision of the methods and prospective interpretation (Reviewer 3). Please respond carefully to these points and I will assess the next revision at desk before issuing a final Stage 1 decision.

Reviewer comments to Author:

Reviewer: 1

Comments to the Author(s)

Thank you very much. All my comments have been addressed sufficiently.

Reviewer: 2

Comments to the Author(s)

The revised manuscript addresses my previous comments. I have one note on a rationale the author gives in their response, though this is not used in the manuscript so it may not warrant further changes.

My previous comment 1 suggested that the author could consider how they would interpret potentially different effects in reaction times and error rates. The manuscript now states that they focus on reaction times, as they are more commonly used in the literature. However their response to my comment also notes:

“Typically, in human information processing models the measures of reaction times and error rates are often seen as measuring the same underlying cognitive process (e.g. Draheim, Hicks, & Engle, 2016).”

I’m not sure this reference supports focusing on reaction times. Draheim et al. propose using a composite measure of RT and accuracy, on the basis that the latency costs typically used in task-switching are contaminated by speed-accuracy trade-offs. The way in which multiple processes contribute to patterns of RT and error effects is also something I have been interested in in my own work using evidence accumulation models (Hedge et al., 2018a, 2018b). For example, in models like the drift-diffusion model, an increase in the boundary separation parameter can lead to an increase in reaction time effects and a decrease (or no observable change) in error rates. This parameter has previously been implicated in conflict adaptation in the form of post-error slowing (Dutilh et al., 2011). Though sequential congruency effects do not follow the same pattern, the point is that basing an interpretation on either reaction times or error rates may overlook theoretically meaningful combinations of the two.

References

Dutilh, G., Vandekerckhove, J., Forstmann, B. U., Keuleers, E., Brysbaert, M., & Wagenmakers, E. J. (2012). Testing theories of post-error slowing. *Attention, Perception, & Psychophysics*, 74(2), 454-465. <https://doi.org/10.3758/s13414-011-0243-2>

Hedge, C., Powell, G., Bompas, A., Vivian-Griffiths, S., & Sumner P. (2018a). Low and variable correlation between reaction time costs and accuracy costs explained by accumulation models: meta-analysis and simulations. *Psychological Bulletin*, 144(11), 1200-1227. <http://dx.doi.org/10.1037/bul0000164>

Hedge, C., Powell, G., & Sumner, P. (2018b). The mapping between transformed reaction time costs and models of processing in aging and cognition. *Psychology and Aging*, 33(7), 1093-1104. <http://dx.doi.org/10.1037/pag0000298>

Reviewer: 3

Comments to the Author(s)

In this second round of reviews, I found a lot of improvement in the submission, especially in the analysis plan section. My big picture view is that I still think that if our own study did not replicate the evidence that the original paper showed for Exp1 then a new study would need more and not fewer participants to further explore this question. Since the present proposal is not planning to use Bayesian analysis, the expected non-significant results won’t allow for any assessment of the original hypothesis. As a registration, I still miss the explicit links between result patterns and interpretations. Below, I explain these and a few other comments.

In their response, the author stated that “the data has already been collected”. I might have got misled on that by the manuscript future tense wording, e.g., “Therefore 100 participants will be collected to achieve sufficient power of .9, given that some participants will need to be excluded..” I find it still confusing.

From our 152 participants, we lost 20 due to exclusions. Online data-collection might be more noisy. I would expect even more exclusions.

Now that the author knows that with a greater sample we could not fully replicate the effect, I wonder if the power analysis would need to be recalculated with our outputs.

I appreciate that an analysis part is added to this registered report. I think that is a crucial part. We can see that the author plans to conduct the main analyses with and without exclusions. It would be important to add how the author would interpret if the results are not in line, which analysis would be the basis of the conclusions.

I cannot seem to find how the sentences will be divided into sentence regions. That was a tough task for us. It would be important to have it worked out before the analysis.

The region-based analysis works with reading time predictions based on the person's reading times. It is crucial to state whether the prediction will be calculated with the inclusion of all the readings, or just the congruent sentences or the filler sentences as well.

I cannot see what pattern of results of the sentence region-based analysis will lead to what conclusions. Again, something crucial for a registration.

It would be great to add 'outcome-neutral analyses', for example to see if there is Stroop effect at all. Without that, the rest of the analyses are not too interesting.

Two experiments described but we don't know how the author would interpret the results if they contradict each other. What pattern of results would replicate the original findings and what would contradict the original hypothesis?

"was presented for xxx ms" is a misleading wording for online experiments. The truth would be something like "We used the xxx method for requesting that the computer present the stimulus for xxx ms" For an explanation, see <https://twitter.com/ceptional/status/1296686833305153536>

Balazs Aczel

Author's Response to Decision Letter for (RSOS-201814.R1)

See Appendix B.

Decision letter (RSOS-210550.R0)

Dear Dr Dudschig

On behalf of the Editor, I am pleased to inform you that your Manuscript RSOS-210550 entitled "Are control processes domain-general? A replication of "To adapt or not to adapt? The question of domain-general cognitive control" (Kan et al., 2013)" has been accepted in principle for publication in Royal Society Open Science.

Please note that you must now register your approved protocol on the Open Science Framework (<https://osf.io/rr>), using the 'Submit your approved Registered Report' option and then the 'Registered Report Protocol Preregistration' option. Please use the Registered Report option even though your article is being accepted as a Stage 1 Replication. Further into the registration process, in the Journal Title field enter 'Royal Society Open Science (Replication article type, Results-Blind track)'. Please note that a time-stamped, independent registration of the protocol is mandatory under journal policy, and manuscripts that do not conform to this requirement cannot be considered at Stage 2. The protocol should be registered unchanged from its current approved state. Please include a URL to the protocol in your Stage 2 manuscript, and because you submitted via the Results-Blind track please note in the manuscript that the pre-registration was performed after data analysis (e.g. 'This article received results-blind in-principle acceptance (IPA) at Royal Society Open Science. Following IPA, the accepted Stage 1 version of the manuscript, not including results and discussion, was preregistered on the OSF (URL). This preregistration was performed after data analysis.')

Please also note that this new registration is required even though you already preregistered your protocol on AsPredicted prior to data collection. Both registrations should be reported at Stage 2.

Following completion of your study, we invite you to resubmit your paper for peer review as a Stage 2 Replication. Please note that your manuscript can still be rejected for publication at Stage 2 if the Editors consider any of the following conditions to be met:

- The Introduction and methods deviated from the approved Stage 1 submission (required).
- The authors' conclusions were not considered justified given the data.

We encourage you to read the complete guidelines for authors concerning Stage 2 submissions at : <https://royalsocietypublishing.org/rsos/replication-studies#AuthorsGuidance>. Please especially note the requirements for data sharing and that withdrawing your manuscript will result in publication of a Withdrawn Registration.

We encourage you to read the complete guidelines for authors concerning Stage 2 submissions at <https://royalsocietypublishing.org/rsos/registered-reports#ReviewerGuideRegRep>. Please especially note the requirements for data sharing and that withdrawing your manuscript will result in publication of a Withdrawn Registration.

Once again, thank you for submitting your manuscript to Royal Society Open Science and I look forward to receiving your Stage 2 submission. If you have any questions at all, please do not hesitate to get in touch. We look forward to hearing from you shortly with the anticipated submission date for your stage two manuscript.

Kind regards,
Professor Chris Chambers
Royal Society Open Science
openscience@royalsociety.org

on behalf of Professor Chris Chambers (Registered Reports Editor, Royal Society Open Science)
openscience@royalsociety.org

Author's Response to Decision Letter for (RSOS-210550.R0)

See Appendix C.

RSOS-210550.R1

Review form: Reviewer 1 (Charlotte Eben)

Is the manuscript scientifically sound in its present form?

Yes

Are the interpretations and conclusions justified by the results?

Yes

Is the language acceptable?

Yes

Do you have any ethical concerns with this paper?

No

Have you any concerns about statistical analyses in this paper?

No

Recommendation?

Accept as is

Comments to the Author(s)

This study presents a very clear replication attempt of Kan et al., (2013). I am impressed by the openness about the analyses and the details provided. The conclusions drawn seem to be appropriate and based on the data.

I just have two comments. One of the comments is with regard to the data and analyses shared: meta data and explanations of your data/variables will help to understand every column of your data and would make your analyses more reproducible and the data more reusable.

Additionally, there seem to be quite some spelling mistakes in the manuscript.

Review form: Reviewer 2

Is the manuscript scientifically sound in its present form?

Yes

Are the interpretations and conclusions justified by the results?

Yes

Is the language acceptable?

Yes

Do you have any ethical concerns with this paper?

No

Have you any concerns about statistical analyses in this paper?

No

Recommendation?

Accept with minor revision

Comments to the Author(s)**Summary**

This is the stage 2 version of a stage 1 replication report that I had previously reviewed. The manuscript attempts to replicate the findings of Experiment 1 from Kan et al. (2013).

Review

I thought this was a well-conducted replication and I have only a handful of minor comments. The analysis seemed to me to be consistent with the plan (though see minor comment in point 1). The authors additionally removed extremely long RTs before analysis, which was not pre-registered but is entirely sensible to me. The interpretation of the results seems well-justified.

- 1) The pre-registered analysis plan states that the results would focus on raw error rates (pg 12, line 5), but the results state that the focus was on the arcsin transformed error rates (pg 14, line 26). It appears to make little difference – the manuscript states that both would be reported if they differed and I didn't see an instance of it - but it is an inconsistency with the analysis plan.
- 2) Wording on page 12 line 3: "Again, here report both analyses are reported..."
- 3) Wording on page 28 line 12: "...participant sample was use based."

Review form: Reviewer 3 (Balazs Aczel)**Is the manuscript scientifically sound in its present form?**

Yes

Are the interpretations and conclusions justified by the results?

No

Is the language acceptable?

Yes

Do you have any ethical concerns with this paper?

No

Have you any concerns about statistical analyses in this paper?

No

Recommendation?

Accept with minor revision

Comments to the Author(s)

I was happy to see that this project reached completion. Although the results are not surprising, I believe that it's important to publish data and results relevant to the investigated question.

I only found minor problems with the submission:

The manuscript has a good number of sentences written in future tense. Searching for "will" or "will be" quickly shows where. Some of these are relevant, but other must be left here from Stage 1.

p.6.

"This preregistration was performed after data analysis." might be a mistake.

p.14

In "For RT, the ANOVA revealed a significant main effect of Stroop congruency with faster responses to congruent trials (654 ms)" and similar places, it must be mentioned that these are mean values.

P.16

"was not significant, $F(1, 89) = 1.46$, $p = .231$, $\eta^2 = 0.02$, indicating that the congruency effect was similar following compatible trials" is an incorrect conclusion as non-significant results cannot indicate such conclusion.

P.20

Please double-check the "difference in both raw and residual reading times (~11 ms difference) within the first critical region, $t_{s(94)} > 4.03$, $p < .001$, $d_z > 0.41$ " results. Appendix table indicates that the SD was 100 ms here, so I was surprised to see such a strong effect for 11ms difference.

Balazs Aczel

Decision letter (RSOS-210550.R1)

Dear Dr Dudschig

On behalf of the Editor, I am pleased to inform you that your Stage 2 Replication submission RSOS-210550.R1 entitled "Are control processes domain-general? A replication of "To adapt or not to adapt? The question of domain-general cognitive control" (Kan et al., 2013)" has been accepted for publication in Royal Society Open Science subject to minor revision in accordance with the referee suggestions. Please find the referees' comments at the end of this email.

The reviewers and Subject Editor have recommended publication, but also suggest some minor revisions to your manuscript. We invite you to respond to the comments and revise your manuscript. Below the referees' and Editors' comments (where applicable) we provide additional requirements. Final acceptance of your manuscript is dependent on these requirements being met. We provide guidance below to help you prepare your revision.

Please submit your revised manuscript and required files (see below) no later than 7 days from today's (ie 27-Apr-2022) date. Note: the ScholarOne system will 'lock' if submission of the revision is attempted 7 or more days after the deadline. If you do not think you will be able to meet this deadline please contact the editorial office immediately.

Please note article processing charges apply to papers accepted for publication in Royal Society Open Science (<https://royalsocietypublishing.org/rsos/charges>). Charges will also apply to papers transferred to the journal from other Royal Society Publishing journals, as well as papers submitted as part of our collaboration with the Royal Society of Chemistry

(<https://royalsocietypublishing.org/rsos/chemistry>). Fee waivers are available but must be requested when you submit your revision (<https://royalsocietypublishing.org/rsos/waivers>).

Kind regards,
Professor Chris Chambers
Royal Society Open Science
openscience@royalsociety.org

on behalf of Professor Chris Chambers (Registered Reports Editor, Royal Society Open Science)
openscience@royalsociety.org

Associate Editor Comments to Author (Professor Chris Chambers):

Comments to the Author:

The three reviewers who assessed the Stage 1 submission kindly returned to evaluate the Stage 2 manuscript. As you will see, all are positive about the completed article and judge that the primary Stage 1 criteria are met. There are some minor issues to attend to in revision, including the curation of the online data, some minor inconsistencies with the original protocol, correct interpretation of non-significant findings, and presentational issues (e.g. concerning tenses and typos). Regarding Reviewer 3's concern with your statement, "This preregistration was performed after data analysis": since you submitted via the results-blind track, your statement is correct.

Provided you are able to respond comprehensively to all points raised in a minor revision and response, final acceptance should be forthcoming without requiring further in-depth review.

Reviewers' comments to Author:

Reviewer: 1

Comments to the Author(s)

This study presents a very clear replication attempt of Kan et al., (2013). I am impressed by the openness about the analyses and the details provided. The conclusions drawn seem to be appropriate and based on the data.

I just have two comments. One of the comments is with regard to the data and analyses shared: meta data and explanations of your data/variables will help to understand every column of your data and would make your analyses more reproducible and the data more reusable. Additionally, there seem to be quite some spelling mistakes in the manuscript.

Reviewer: 2

Comments to the Author(s)

Summary

This is the stage 2 version of a stage 1 replication report that I had previously reviewed. The manuscript attempts to replicate the findings of Experiment 1 from Kan et al. (2013).

Review

I thought this was a well-conducted replication and I have only a handful of minor comments. The analysis seemed to me to be consistent with the plan (though see minor comment in point 1). The authors additionally removed extremely long RTs before analysis, which was not pre-registered but is entirely sensible to me. The interpretation of the results seems well-justified.

- 1) The pre-registered analysis plan states that the results would focus on raw error rates (pg 12, line 5), but the results state that the focus was on the arcsin transformed error rates (pg 14, line 26). It appears to make little difference – the manuscript states that both would be reported if they differed and I didn't see an instance of it - but it is an inconsistency with the analysis plan.
- 2) Wording on page 12 line 3: "Again, here report both analyses are reported..."
- 3) Wording on page 28 line 12: "...participant sample was use based."

Reviewer: 3

Comments to the Author(s)

I was happy to see that this project reached completion. Although the results are not surprising, I believe that it's important to publish data and results relevant to the investigated question.

I only found minor problems with the submission:

The manuscript has a good number of sentences written in future tense. Searching for "will" or "will be" quickly shows where. Some of these are relevant, but other must be left here from Stage 1.

p.6.

"This preregistration was performed after data analysis." might be a mistake.

p.14

In "For RT, the ANOVA revealed a significant main effect of Stroop congruency with faster responses to congruent trials (654 ms)" and similar places, it must be mentioned that these are mean values.

P.16

"was not significant, $F(1, 89) = 1.46$, $p = .231$, $\eta^2 = 0.02$, indicating that the congruency effect was similar following compatible trials" is an incorrect conclusion as non-significant results cannot indicate such conclusion.

P.20

Please double-check the "difference in both raw and residual reading times (~11 ms difference) within the first critical region, $t_s(94) > 4.03$, $p_s < .001$, $d_z > 0.41$ " results. Appendix table indicates that the SD was 100 ms here, so I was surprised to see such a strong effect for 11ms difference.

Balazs Aczel

===PREPARING YOUR MANUSCRIPT===

one version should clearly identify all the changes that have been made (for instance, in coloured highlight, in bold text, or tracked changes);

===PREPARING YOUR REVISION IN SCHOLARONE===

- If you are providing image files for potential cover images, please upload these at this step, and inform the editorial office you have done so. You must hold the copyright to any image provided.
- A copy of your point-by-point response to referees and Editors. This will expedite the preparation of your proof.

- Ensure that your data access statement meets the requirements at <https://royalsociety.org/journals/authors/author-guidelines/#data>. You should ensure that you cite the dataset in your reference list. If you have deposited data etc in the Dryad repository, please only include the 'For publication' link at this stage. You should remove the 'For review' link.
- If you are requesting an article processing charge waiver, you must select the relevant waiver option (if requesting a discretionary waiver, the form should have been uploaded, see 'File upload' above).
- If you have uploaded any electronic supplementary (ESM) files, please ensure you follow the guidance at <https://royalsociety.org/journals/authors/author-guidelines/#supplementary-material> to include a suitable title and informative caption. An example of appropriate titling and captioning may be found at https://figshare.com/articles/Table_S2_from_Is_there_a_trade-off_between_peak_performance_and_performance_breadth_across_temperatures_for_aerobic_scope_in_teleost_fishes_/3843624.

Author's Response to Decision Letter for (RSOS-210550.R1)

See Appendix D.

Decision letter (RSOS-210550.R2)

Dear Dr Dudschig:

I am pleased to inform you that your manuscript entitled "Are control processes domain-general? A replication of "To adapt or not to adapt? The question of domain-general cognitive control" (Kan et al., 2013)" is now accepted for publication in Royal Society Open Science.

Please remember to make any data sets or code libraries 'live' prior to publication, and update any links as needed when you receive a proof to check - for instance, from a private 'for review' URL to a publicly accessible 'for publication' URL. It is also good practice to add data sets, code and other digital materials to your reference list.

Royal Society Open Science is a fully open access journal. A payment may be due before your article is published. Our partner Copyright Clearance Center's RightsLink for Scientific Communications will contact the corresponding author about your open access options from the email domain @copyright.com (if you have any queries regarding fees, please see <https://royalsocietypublishing.org/rsos/charges> or contact authorfees@royalsociety.org).

on behalf of Professor Chris Chambers (Subject Editor).

Follow Royal Society Publishing on Twitter: @RSocPublishing
Follow Royal Society Publishing on Facebook:
<https://www.facebook.com/RoyalSocietyPublishing/>
Read Royal Society Publishing's blog:
<https://royalsociety.org/blog/blogsearchpage/?category=Publishing>

Appendix A

Reviewer: 1

Comments to the Author(s)

The author proposes a study to directly and conceptually replicate the Experiment 1 by Kan et al. (2013). In my opinion this is a sound replication proposal and I am impressed by the details the author provides especially with regard to the online testing procedure.

I have only a few minor suggestions you might want to consider:

- In my opinion there are details about the statistical model, the factors and levels and the inference criteria missing in the data analysis plan but also in the pre-registration.

I have added more details regarding the planned analysis and the statistical tests.

- For me the theoretical motivation for the conceptual replication (Experiment 2) is not entirely clear.

I have now added this motivation to the manuscript at the start of Experiment 2. The core point lies in the well-studied increase of the Stroop conflict if using response-based conflict, which is due to two mechanisms typically contributing to the Stroop effect – a semantic conflict and a response competition mechanism. If the conflict observed within the stimulus-based Stroop task is small, the opportunity to observe a reduced Stroop effect following difficult to process sentences is also reduced. Additionally, I wanted to keep Experiment 1 and Experiment 2 as identical as possible with the exception of the Stroop materials.

- Has the congruent/incongruent procedure been tested online? I am a huge fan of online testing but from my experience I would advise to run a short pilot to see if you can find the effect in the sentence reading task.

Yes, I agree here. I have experience using online data collection procedures and have found the results to be relatively consistent with lab-based testing. Although this testing has been with more standard reaction time tasks, I have some data from other online experiments using the moving window procedure showing robust effects. Of course, one aspect that previous experience has taught me is that participant exclusion tends to be a bit higher in online than in lab-based studies. Here, the use of predefined data quality exclusion criterion is essential. For example, ten of the filler materials used in the sentence task are followed by comprehension questions. I will remove participants who did not perform at an adequate level to these questions (a criterion used in the original study) and importantly will perform additional analyses (see manuscript, follow-up analyses) which further ensure data quality beyond data-quality-checking standards from the original paper.

- The original study does not have a big sample size, it might be possible that the original effect size is overestimated due to a lack of power in the original study. Maybe this is something to keep in the back of your mind?

Yes, I agree here. The original sample size was small but the replication will more than double the original number of participants. In addition, as Experiment 2 is essentially the same experiment, it is possible to combine for an even larger sample size (I will report such a combined analysis in additional section now).

- In my experience you might encounter some technical difficulties while data collection as well. What do you plan on doing with data sets where single trials are missing (e.g. due to instable

internet connection)? What will you do with participants that start the experiment again because they might have accidentally closed the window?

Only complete data sets will be analysed. The data is stored on our local server at the end of the experiment. I have not had any issues with lost trials within a single data file across several large samples of participants so far, so I don't anticipate this to be a big problem. From my experience using MTurk, participants cannot re-accept a task once they have started it. Also, given that we pay participants a comparably high rate, the tasks are typically rather popular and the link expires quickly, not leaving many options to return to the task.

- Maybe consider using Prolific instead of Mturk? Mturk is not made specifically for academic research whereas Prolific is (Palan & Schitter, 2018; <https://doi.org/10.1016/j.>). Of course, this is not at all required, I just have very good experience with Prolific and the way they try to ensure data quality.

Currently, I have more experience using MTurk and therefore run that study on MTurk. However, I have looked at Prolific and I think I will consider that as an option in the future.

- De Leuw (2015) himself suggests using Chrome as this is best tested in jsPsych, it also seems that Chrome (in combination with Windows) “adds the least random noise to RT measurement” (see Pronk et al. 2020; <https://doi.org/10.3758/>).

I do not require that participants use a specific OS/browser combination. Pilot testing did ensure the experiment runs on Windows/Mac/Linux using Firefox/Safari + Chromium based browsers. I only specify that users should avoid Internet Explorer. I think that although precision within online studies not always matches that of a lab based environment (e.g., I would not be keen to run a low-level perception psychophysics experiments within an online environment, nor experiments involving the synchronisation of tone and visual stimuli), online studies can achieve good enough precision to be useful in the current setup. Regarding restricting participants to only Windows/Chrome, I fell that this is not necessary. For example, the precision difference across different OS/browser combinations reported in Bridges et al. (2020) did not seem to indicate substantial differences large enough to require participants to use a specific OS/browser combination with jsPsych. However, thanks for pointing this out to me, and I definitely will keep this in mind for future studies. Given that the paper has been submitted at the result-blind track to the journal, I cannot change anything regarding data collection now. But I did implement several data check options in the paper, that allow to access whether the measurements worked with regard to all basic effects (Stroop effect, conflict adaptation within the Stroop task etc.)

- The combination of data sets as described in the pre-registration might be interesting here as well.

Yes, I agree here, I will now report this analysis.

- In the spirit of open science and to make it easier to follow your analyses pipeline you could also share the code and the analyses scripts together with the pre-registration or at the later stage together with the data.

Yes, all data files and analysis scripts will be uploaded to an appropriate repository. The study was registered at “As Predicted”. I plan to upload the data to ZENODO, but of course also OSF if required by the journal or reviewer.

And some very minor points:

- On p. 7 in line 20/21 it has to be Braem instead of Bream.

Fixed

- On p. 10 line 22: do you mean “right” finger or ring finger? Because for left-handed people it is the left finger?

Fixed

Reviewer: 2

Comments to the Author(s)

Summary

The manuscript is a stage 1 replication study of the paper:

Kan, I. P., Teubner-Rhodes, S., Drummey, A. B., Nutile, L., Krupa, L., & Novick, J. M. (2013). To adapt or not to adapt: The question of domain-general cognitive control. *Cognition*, 129(3), 637-651.

The author proposes to replicate Experiment 1 from Kan et al., as well as conducting a conceptual replication using a small modification. Experiment 1 of Kan et al. had participants perform intermixed trials of a sentence comprehension and Stroop task. The observation of a two-way interaction between preceding (sentence) congruency and Stroop task congruency was interpreted as evidence for domain general conflict adaptation – the Stroop effect was smaller when it followed an ambiguous sentence. The original Experiment 1 used Stroop stimuli where the incongruent words (‘brown’, ‘orange’, ‘red’) did not match the font colours to be named (‘blue’, ‘green’, ‘yellow’). The conceptual replication proposed here (Experiment 2) modifies this so that they do overlap.

Review

I found the proposal to be generally clear and well-written. I thought that the primary and secondary criteria were met, or nearly met with a potential small elaboration (point 1). The proposed replication appears to match the details of the original experiment, with the perhaps notable exception that it will be conducted with an online sample. I was satisfied with the author’s justification for why this does not compromise the ability of the proposed study to detect the effect of interest. The author proposes numerous secondary analyses which I thought were sufficient quality control checks. I have a couple of minor comments on the logic of the conceptual replication study, but they are probably not critical.

1) I thought that there could be a more explicit statement about the critical hypothesis test(s), as it may not be obvious to a reader not familiar with Kan et al. From the power analysis, I assume that the main test is the two-way interaction between previous-trial and current-trial congruency. Some elaboration may be warranted on whether there is an expected direction/pattern of condition means. Further, the author proposes to analyse RTs, error rates, and arcsin transformed error rates (as in the original). With multiple outcomes, there is potential flexibility in determining whether the original findings replicated. It might be worth commenting on whether significant effects in one or all outcomes will be considered as sufficient evidence.

Thanks for pointing this out to me. It is indeed the fact that in the original Kan paper there were two ANOVAs mentioned for the error rates (raw proportions and an arcsin transformation) but also for the RT data there were two analyses mentioned (one excluding outliers and one replacing the outliers). However, for each dependent variable, the authors report that the results across all analyses showed consistent patterns, whereby they chose to only report the details for one of these analyses for the error rates and RTs.

With regard to the question what conditions need to be fulfilled to call it a successful replication, your comment was very helpful to think about this more clearly. Typically, in human information processing models the measures of reaction times and error rates are often seen as measuring the same underlying cognitive process (e.g. Draheim, Hicks, & Engle, 2016). Nevertheless, there is a strong tradition to focus more heavily on the reaction time patterns.

Therefore, I focus on the two-way interaction between previous sentence congruency and current Stroop congruency in reaction time. In line with the conflict adaptation effect, it is predicted that the conflict effect is reduced following incongruent sentences (= direction / pattern of means). From the background of the conflict monitoring literature, and specifically the empirical evidence demonstrating these conflict adaptation effects, the effects – if existent - are basically almost always present in the reaction time data (and if reported, typically mirrored in the error data). In an extreme case, participants could focus on accuracy in the current study and commit hardly any errors. Therefore, my assessment of whether the effect replicates or not will be biased towards the data pattern observed in reaction time. I state all these things more clearly in the manuscript now.

Draheim, C., Hicks, K. L., & Engle, R. W. (2016). Combining reaction time and accuracy: The relationship between working memory capacity and task switching as a case example. *Perspectives on Psychological Science*, 11, 133–155.

2) The conceptual replication seems sensible on the surface, though it did raise a couple of questions in my mind that the author might want to consider.

The reasoning given is that Experiment 2 “...should result in larger conflict... and potentially larger cross-task transfer effects”. One interpretation of this is that conflict needs to be sufficiently large for transfer to occur. Another is that transfer could occur in both experiments, but the effect size would be larger in Experiment 2. If it is the latter, what is the benefit of the additional Experiment over increasing the power of Experiment 1 for a smaller effect size? I don’t object to the planned sample size, and it is already substantially larger than the original study. However, my reading is that it is based on the original effect size, and prior replication efforts have indicated that it is common for replications to produce smaller effect sizes than the original studies (e.g. Open Science Collaboration, 2015).

As we use the same set of sentences in both experiments, the size of conflict experienced from sentences to Stroop trials should be similar (which is the critical effect reported by Kan et al.). However, in order to observe a difference in the size of conflict experienced in the Stroop trials depending on previous sentence congruency, I wanted to increase the size of conflict effect specifically within the Stroop trials. I still agree with the reviewer, that another option would be to increase the sample size of Experiment 1. However, as also noted by reviewer 1, the high similarity between Experiment 1 and 2 does not exclude the possibility for a combined analysis. Indeed, I have now added a section to the paper where such a combined analysis will be conducted to deal with potential issues of power due to the true effect size potentially being smaller than the one originally reported in the study of Kan et al.

3) Kan et al. make the distinction between representational conflict and response conflict (Egner, 2008), and focus on representational conflict throughout their experiments. In their Stroop tasks, and in Experiment 1 here, the incongruent words were not part of the eligible response set. Experiment 2 here aims to elicit a larger conflict effect by adding response conflict (the incongruent

words are part of the eligible response set). The author may want to comment on whether response conflict should also be expected to contribute to the transfer effect. Naively, I have no reason to expect that it should not, but the prediction doesn't necessarily follow from the original study.

I agree that the original paper focuses on representational conflict. Importantly, the Stroop conflict used in Experiment 2 in my conceptual replication contains both response and stimulus (i.e. representational) conflict. Given that the representational conflict in the Stroop task originates on a semantic level, and the sentence conflict originates on a syntactic level, I don't see a reason for the conflict transfer effect to be limited to this specific Stroop conflict type. I have now clarified this in more detail within the manuscript.

4) Just to bring it to the author's attention if they are not already aware - there is preprint available online of another proposal to replicate Kan et al.: <https://psyarxiv.com/5k8rq/>

Thanks for pointing this out, I am now aware of the existing replication (see Reviewer 3 being an author of the other replication) and cite it in the paper.

Reviewer: 3

Comments to the Author(s)

Before going into details of the review, I have to mention that I am in a special situation as I have been requested to be a reviewer of this submission while I have a Stage 2 submission under review at RSOS. In fact, my team has replicated the very same study, Kan et al., 2013.

An advantage of the registered report format that we can give advice to the submitters before too much investment in the project has been done. Saying that I would discourage the author to conduct the replication, or in the proposed form for the following reasons:

We replicated Exp 1 from Kan et al. in three countries by independent labs with greater sample size. Compared to the proposed work, we very closely followed the original protocol and conducted the experiments in the lab and not online (some other deviations of the proposal are discussed below).

Our results were rather discouraging as you can read it in our preprint: <https://psyarxiv.com/5k8rq>

Thanks for pointing this out, I now cite that there is another replication study and that is failed to replicate the core findings of Kan et al. (2013).

Besides all of this, even the original study of Kan et al. (2013) raised doubts that Exp. 1 is a good test of the theory, that is why they (and we) conducted Exp. 2 and 3. but the submission doesn't propose to replicate them. Nevertheless, our results were similarly pessimistic about the presence of congruency sequence effect, making us believe that the cross-task congruency sequence effect is either nonexistent or the design of Kan et al. is not a good test of the effect.

In short, I would encourage the author to use a different design to test to effect. My lab is open for further discussion about the potential empirical directions.

Response: Thanks for all these suggestions, I chose to replicate specifically Experiment 1 with a sample size resulting in sufficient power as my research is specifically interested in linguistic conflict and how linguistic conflict interacts with other types of conflict processing. As the other reviewers pointed out – the changes between my two experiments are so small – that is also an option to combine the analysis to see whether there really is an underlying power issue. I added this

planned analysis now to the manuscript. It would be great to be in contact about potential further directions – thanks very much for that option – however the current paper has been submitted on the result-blind track to the journal. This means that the data has already been collected before this submission, but the outcome is not revealed to the reviewers (which is an option given by the journal). I read your replication, it is really interesting, I have also learned about a few things, which seemed not reported in the original Cognition paper (more details below). I just wrote down what stood out to me, but there might be some points interesting to discuss.

Below, I list a few more observations about the proposed design.

Is the proposed Exp1 a direct replication?

I think there are a few deviations from the original study. I do not mind them saying that it is a direct replication but it should be apparent that there are deviations.

Online vs in lab

Thanks for careful reading and assessment of the submitted replication. I have now highlighted any potential deviation between the current replication and the original study, and your comments were really helpful there. The question of whether this is a direct replication or not is difficult. I believe that all deviations from the original article are small (except for online vs. Lab) and are now clearly stated in the revision. What was very surprising for me was the following: In the process of revising the current re-submission and careful reading of the replication of your group, I came to the conclusion that there must be slight issues with the description of the original study. I will outline these details below, but the only reasonable way for me to explain the discrepancies between your replication and the Kan paper method section would be that Kan et al. didn't report precisely what materials were used within the baseline Stroop task. I also don't think that this is a major issue, but it is very surprising.

Therefore, it is questionable whether the original study and the pre-print replication followed the details reported in the respective method sections within the manuscripts (see comments below). I suspect that the authors of the pre-print used the materials sent by Kan, which weren't reported to all final details in the paper. Anyway, I don't think this is an issue for the effect we are interested in. But I see for example one benefit of my online data collection: I think the advantage of using the online format might also be that it was possible for me to test US native speakers, therefore regarding the language background a very similar sample to the original study, which is not the case in the other replication (testing people in Australia, Singapore, UK). I now didn't check whether you adjusted your items to British English spelling, but in both cases this would be a deviation from the original study. I know this soon becomes a philosophical question what a direct replication is, but in summary I don't think that the differences in my replication are that much more critical than in the other replication.

“The mapping of keys (G, H, J) is randomly assigned to a response colour (blue, green, yellow) for each individual participant,...” The original study doesn't randomize it.

Yes, that is a difference. However, it is a very uncommon procedure not to randomize assignments of stimuli to response keys. If this is the driving force behind the Kan study effects, it would be very surprising. I now further highlight this discrepancy.

Page 10 line 59 (footnote): I believe this is an important deviation from the original study, however, I am not sure I understand how the stimulus set looked like. There were 42 non-filler sentences in the original study. The footnote suggests that participants did not see all the 42 sentences but just one pseudo-randomized set of 21 sentences. In the original study, the participants saw all the 42

sentences. As the materials for Experiment 1 of Kan et al. 2013 are available the specific order of the sentences in the original study could be applied in the replication as well.

This is a misunderstanding caused by the footnote. I now write this more clearly. Participants do see 42 non-filler items (21 for congruent/21 from incongruent sentences) in the current replication, which is in line with the original study. I do present them in a randomised order (constrained such that a non-filler sentence is followed by a Stroop trial). I think the requirement that they are presented in the same specific order as the original study is questionable. Indeed, it is not clear from the original study that a fixed random order was used for all participants. I did this as again it is a more standard procedure to randomize materials and sentence order instead of keeping that identical across all participants. I used a randomization procedure that ensured that each critical sentence was followed by a Stroop task (so no critical trials are lost that are relevant to test the hypothesis). Indeed I think the fixed order of items across all participants within the original study is actually questionable (resulting in potential confounds of specific order).

Again, I now highlight this discrepancy regarding the randomization more clearly in the paper.

Page 11 line 26: I am not sure that I understand the reasoning behind the exclusion criteria on the comprehension probe task. The author mentions Kan excluding only one participant at a chance level (50%) and that all the other participants score above 70%. The author then states that the replication will follow Kan et al. (2013) and participants with 6 or fewer correct responses will be dropped. This indicates a 60% (given that there were 10 comprehension probe sentences) cutoff level which is neither 50% nor 70%.

I do not understand this point. I believe this is the same comprehension criterion as the original paper. If I remove participants with 6 or fewer correct responses, the accuracy rate of remaining participants will be at least 70%. From Kan et. al "The remaining participants ($n = 39$) scored at or above 70% ($M = .90$, $SD = .09$)". I clarified this in the paper.

The whole sentence preprocessing part is missing. This is not a problem in itself but should be noted as a deviation and justified.

I plan to analyse the sentence task following the procedure described in Kan et al. Specifically, raw reading times 2.5SDs beyond the subjects mean across all conditions are replaced with the 2.5SD cutoff value. Subsequently, regression will be used to predict raw reading time from sentence region length (number of characters). Each individual's predicted reading time is then subtracted from their actual reading time to give a residual reading time. T-tests are then performed on the residual reading time in each sentence region, most importantly, within the temporarily ambiguous and disambiguating regions. I now clarify this. Thanks for pointing this out. I did not have it in the original submission, as the focus was on the transfer effect to the Stroop trials.

Page 10 line 52: The test part consists of 197 trials. In Kan they used 60 cong stroop + 60 incong stroop + 21 cong and 21 incong sentences which is 162. If the filler sentences are included that is $162 + 29 = 191$.

This is correct. My test phase will consist of 63 congruent Stroop and 63 incongruent Stroop stimuli. This change was made in order to balance the possible combinations of relevant and irrelevant dimensions within the Stroop task. Interestingly, this is not something that seems to be balanced within the original study. For example, within subjects the word "red" did not appear in blue the same number of times as it did orange. Whilst this is not critical to the hypothesis under investigation, it is the reason for my slight change in the number of Stroop stimuli presented. This is

not something that is possible to ascertain from either the method section of the original study. I also further highlight this discrepancy.

In addition, I think it is also relevant to highlight another difference between my replication and the original Kan et al. and the pre-print replication manuscripts. Again, I do not think this aspect of the experiment was clear in either the original study nor the replication, and only became evident when looking at the results of the pre-print replication (NB. I do not know for certain if this was the case for the original Kan paper). The baseline "Stroop" task contained non Stroop-like stimuli items, such as the words tax, sounds, hungry. In total 30 words were presented, of which 6 were colour words. Thus, the reported numbers of congruent vs. incongruent trials within the baseline Stroop blocks are incorrect. Specifically, from Kan "Then they completed a baseline block of 145 Stroop trials (intermixed congruent and incongruent)" and from pre-print replication "It was followed by a baseline trial with 145 intermixed congruent and incongruent Stroop stimuli." I do not think this is correct, as the actual baseline block contained 61 congruent and 60 incongruent stimuli, with 24 items being non Stroop-like items. My baseline block contains only stroop-like items with 72 in each congruency condition. Again, I do not think this overly problematic regarding the hypotheses tested, but I think it should be made clear within the method section. Why this was not clearly indicated in the original paper and the pre-print replication is unclear.

Page 11 line 59 (footnote): The stepwise reduction and testing for the presence of the Stroop effect are problematic if NHST is used to test the presence of the effect. Continuous testing can modify the alpha level hence the results.

I agree this is potentially an issue. However, it is not really a step-wise reduction of the 2x2 interaction of interest, but rather of participants who do not show a sentence congruency effect, nor a clear Stroop conflict adaptation effect in the baseline. This is only used as an additional test of the 2x2 interaction; specifically, if the 2x2 interaction is absent even within a subset of participants that do show both a clear sentence congruency effect and a clear Stroop to Stroop adaptation effect within the baseline, the evidence against cross-task adaptation would be rather strong (as long as the power remains sufficient). I have now more clearly stated the critical tests in line with the original study, and that these step-wise-tests are really just seen as an additional source of information to strengthen the conclusions.

General comments

Page 10 line 55: The started sentence is not finished.

Thanks, now corrected.

Page 11 line 15: The footnote mark could be moved to the sentence regarding the RT outlier exclusions.

I moved the footnote and have now clarified the whole analysis section to improve the structure.

Page 7 line 32: I am not sure that I am convinced that Exp 2 is needed based on this one-sentence explanation.

I have now clarified the motivation of experiment 2 at the start of Experiment 2.

Page 7 line 55: What was the exact effect size used for the sample size determination?

*The effect-size was calculated from the 2*2 interaction of previous Sentence congruency and current Stroop congruency on RT. The number of required participants was calculated from the partial eta squared value here, and using the R package Superpower. I clarified this in the paper.*

Lakens, D., & Caldwell, A. R. (2019). Simulation-based power-analysis for factorial ANOVA designs. *PsyArxiv*.

There is nothing about the analysis procedure in the paper. Nor about the inferences that the authors will make based on the results. Again, a lot of questions remain unanswered.

Thanks for pointing this out. The paper now includes these details. The planned analysis is, in a first step, identical to the original study, with the critical tests being on the reaction times. The revised version makes now clear what results will be interpreted as a successful replication.

Appendix B

The reviewers and handling editors have recommended publication, but also suggest some minor revisions to your manuscript. Therefore, I invite you to respond to the comments and revise your manuscript.

Associate Editor Comments to Author (Professor Chris Chambers):

The Stage 1 manuscript was returned to three reviewers who assessed the original submission. The reviews are broadly positive and we are now closer to IPA. Reviewer 1 is now satisfied but there remain some final issues to address concerning the consideration of RTs and error rates (Reviewer 2), and clarity and precision of the methods and prospective interpretation (Reviewer 3). Please respond carefully to these points and I will assess the next revision at desk before issuing a final Stage 1 decision.

Reviewer comments to Author:

Reviewer: 1
Comments to the Author(s)

Thank you very much. All my comments have been addressed sufficiently.

Response: Thanks!

Reviewer: 2
Comments to the Author(s)

The revised manuscript addresses my previous comments. I have one note on a rationale the author gives in their response, though this is not used in the manuscript so it may not warrant further changes.

My previous comment 1 suggested that the author could consider how they would interpret potentially different effects in reaction times and error rates. The manuscript now states that they focus on reaction times, as they are more commonly used in the literature. However their response to my comment also notes:

“Typically, in human information processing models the measures of reaction times and error rates are often seen as measuring the same underlying cognitive process (e.g. Draheim, Hicks, & Engle, 2016).”

I'm not sure this reference supports focusing on reaction times. Draheim et al. propose using a composite measure of RT and accuracy, on the basis that the latency costs typically used in task-switching are contaminated by speed-accuracy trade-offs. The way in which multiple processes contribute to patterns of RT and error effects is also something I have been interested in in my own work using evidence accumulation models (Hedge et al., 2018a, 2018b). For example, in models like the drift-diffusion model, an increase in the boundary separation parameter can lead to an increase in reaction time effects and a decrease (or no observable change) in error rates. This parameter has previously been implicated in conflict adaptation in the form of post-error slowing (Dutilh et al., 2011). Though sequential congruency effects do not follow the same pattern, the point is that basing an interpretation on either reaction times or error rates may overlook theoretically meaningful combinations of the two.

References

- Dutilh, G., Vandekerckhove, J., Forstmann, B. U., Keuleers, E., Brysbaert, M., & Wagenmakers, E. J. (2012). Testing theories of post-error slowing. *Attention, Perception, & Psychophysics*, 74(2), 454-465. <https://doi.org/10.3758/s13414-011-0243-2>
- Hedge, C., Powell, G., Bompas, A., Vivian-Griffiths, S., & Sumner P. (2018a). Low and variable correlation between reaction time costs and accuracy costs explained by accumulation models: meta-analysis and simulations. *Psychological Bulletin*, 144(11), 1200-1227. <http://dx.doi.org/10.1037/bul0000164>
- Hedge, C., Powell, G., & Sumner, P. (2018b). The mapping between transformed reaction time costs and models of processing in aging and cognition. *Psychology and Aging*, 33(7), 1093-1104. <http://dx.doi.org/10.1037/pag0000298>

Response: Thanks for pointing these issues and papers out to me. I fully agree that the interpretation of reaction times and error rate results is not conclusive, especially if there are diverging findings. As the current paper is a replication of the original Kan et al. paper – which strongly focuses on the interpretation of the reaction time data – the present paper will pursue a similar approach as the original paper. However, if there are diverging findings, I will use the above suggested approaches for additional discussions and for corroborating the conclusions drawn. I know also acknowledge in the paper that I might oversee a potential theoretical meaningful finding by not using a combined approach: “However, I acknowledge that such an approach might oversee potential meaningful theoretical combinations of the two measures (Hedge, Powell, Bompas, Vivian-Griffiths, & Sumner, 2018a; Hedge, Powell, & Sumner, 2018b).”

Reviewer: 3

Comments to the Author(s)

In this second round of reviews, I found a lot of improvement in the submission, especially in the analysis plan section. My big picture view is that I still think that if our own study did not replicate the evidence that the original paper showed for Exp1 then a new study would need more and not fewer participants to further explore this question. Since the present proposal is not planning to use Bayesian analysis, the expected non-significant results won't allow for any assessment of the original hypothesis. As a registration, I still miss the explicit links between result patterns and interpretations. Below, I explain these and a few other comments.

In their response, the author stated that “the data has already been collected”. I might have got misled on that by the manuscript future tense wording, e.g., “Therefore 100 participants will be collected to achieve sufficient power of .9, given that some participants will need to be excluded..” I find it still confusing.

Response: Thanks, the writing was now adjusted.

From our 152 participants, we lost 20 due to exclusions. Online data-collection might be more

noisy. I would expect even more exclusions.

Now that the author knows that with a greater sample we could not fully replicate the effect, I wonder if the power analysis would need to be recalculated with our outputs.

Response: Our power analysis was calculated on the original study and was also pre-registered. In addition, we have the possibility to conduct a combined analysis across our two very similar experiments giving an approximate sample size of 200. I agree that a higher number of exclusions is a possibility with online data-collection, however, from several other studies (with other paradigms) the dropouts were often not much bigger than in lab studies. If there would be an extensive issue dropping below the required power, this will be recognized and dealt with appropriately. Also, the combination of the experiments is a potential approach to address any potential power issues. This information is now added to the manuscript.

I appreciate that an analysis part is added to this registered report. I think that is a crucial part. We can see that the author plans to conduct the main analyses with and without exclusions. It would be important to add how the author would interpret if the results are not in line, which analysis would be the basis of the conclusions.

Response: This double analysis was planned as it was also reported in the original study. In the original study, the authors reported identical evidence in both measures; therefore, I expect to find converging results in both measures to name it a successful replication. This point can already be found in the manuscript: "Thus, here both ANOVAS are reported and also expected to show similar patterns for a successful replication."

I cannot seem to find how the sentences will be divided into sentence regions. That was a tough task for us. It would be important to have it worked out before the analysis. The region-based analysis works with reading time predictions based on the person's reading times. It is crucial to state whether the prediction will be calculated with the inclusion of all the readings, or just the congruent sentences or the filler sentences as well. I cannot see what pattern of results of the sentence region-based analysis will lead to what conclusions. Again, something crucial for a registration.

Response: Yes, I agree that the precise sentence analysis is not particularly clear from the original Kan et al. (2013) paper. I intend to split the sentences into regions. For example, an incongruent trial can have an initial region, an ambiguous region, a disambiguating region, and a final region. I think it is also important to state that the measure of slower reading times in ambiguous sentences is not the crucial test for the central hypothesis (that it is rather the conflict adaptation effects measured on the Stroop trials). I've now specified this in the manuscript (see sentence analysis section).

The frustrated tourists understood | that | the message | would mean | they couldn't go. (Congruent)
The frustrated tourists understood | | the message | would mean | they couldn't go. (Incongruent)

Here, the critical comparisons concern both the ambiguous region (red) and the disambiguating region (green). Like Kan et al. (2013), this analysis will involve t-tests on both the raw and residual reading times, adjusted for word length and the number of words within a region. Of course, a single participant did not see the identical item in both its congruent and incongruent form, with each version presented in separate lists across participants. (Side note: Interestingly, it was unclear to me if this was the case within the Stage 2 replication by the reviewer. For example, from a quick look at the output files, I did not see the sentence "The frustrated tourists understood that the message would mean they couldn't go."). Regarding a participant's predicted reading times, these

will be based on all sentences. Regarding the conclusions from the sentence reading analysis, it is critical that there is a significant difference between our reading measures within the congruent and incongruent sentences. Without this, it is difficult to argue that the participants experienced conflict, and thus, showed subsequent adaptation to the Stroop conflict. All these issues are dealt with in the analysis section of the manuscript (see also section follow-up analysis). With regard to the conclusions: The region-based analysis method in the sentence reading part is preliminary there to test whether participants experienced conflict during the sentence reading processes and therefore serves as a basis for the experiment (to measure the crucial conflict-adaptation effects in the Stroop task).

As mentioned above, the critical test for the hypothesis is performed on the Stroop trials. At this point, I believe it is important to mention a difference in the outlier procedure that I intend to use and the procedure used within the Stage 2 replication by the reviewer (NB. It is unclear to me exactly what was implemented within the original Kan et al. (2013) paper). Specifically, I will replace all RTs falling beyond 2.5 SD's of the participant's mean with the 2.5 SD threshold value. For the critical comparisons (i.e., the inter-mixed Stroop trials that follow sentence items), the values will be based on the all Stroop trials within that block, not just the subset of Stroop trials following the 42 sentence items. Looking at the analysis of the Stage 2 replication, I see that the cutoff values were determined using only the Stroop trials following the sentence items. I think this can be problematic in some cases when a low number of trials remain. For example, one participant (participant_id = 1052) actually has a negative low cutoff value and as a result has RTs (e.g., 51, 53, 79 ms) that are not classified as RT outliers. I think most studies employing reaction time measures would consider responses faster than 100 ms (especially within a three-choice task setting) as being invalid responses. I have clarified this in my manuscript. It was interesting to have a close look at the other replication study's procedures and outcomes.

It would be great to add 'outcome-neutral analyses', for example to see if there is Stroop effect at all. Without that, the rest of the analyses are not too interesting.

Response: Yes, these issues are taken care of in the follow-up analysis section.

Two experiments described but we don't know how the author would interpret the results if they contradict each other. What pattern of results would replicate the original findings and what would contradict the original hypothesis?

Response: If the effect of interest replicated across all experiments in all measures this would be regarded strong evidence in favor of the original effect. If the crucial effect does not show in any of the experiment and measurements, this would be regarded as strong evidence that the effect doesn't exist. If there is a mixture of results in the single experiments, this will be regarded as weak evidence for the original effect and in this case the combined analysis of both experiments will be considered most strongly with regard to the question whether the replication was successful or not. This information has now been added to the manuscript. "As the present study reports two very similar experiments, it is expected that both experiments show conflict-adaptation patterns transferring from the sentence reading task to the Stroop task if domain-general conflict monitoring effects are active. However, if the findings diverge (or unexpected power issues occur due to large exclusion rates), the main interpretation will be based in the combined analysis of both experiments."

"was presented for xxx ms" is a misleading wording for online experiments. The truth would be something like "We used the xxx method for requesting that the computer present the stimulus for xxx ms" For an explanation, see <https://twitter.com/ceptional/status/1296686833305153536>

Response: Thanks for the Twitter reference. I am not sure if this objection is specific to online experiments or timing procedures in general. From reading the thread, it seems the author objects to the use of online studies. However, timing issues exist also in lab-based studies, as not all in-lab software correctly reports missed flips. My reading of the Bridges et al. (2020) paper was that - in general - online data-collection methods can provide valid reaction time data (I acknowledge that the authors of this paper are heavily involved with PsychoPy/PschoJS and the pay-per-use Pavlovia server associated with PsychoJS).

“I have encountered very high variability in the presentation time of dynamic stimuli. For example, an attentional blink program online was found to present stimuli about 30 times slower in one web browser”

I am slightly sceptical about this claim and perhaps have misunderstood what is meant by “30 times slower”. I cannot find any peer-reviewed article showing such large timing variability. Indeed, at a minimum, if one frame is requested, assuming a 60Hz (most common refresh rate), this would be almost a 500 ms difference. If this refers to a variance measure (30 times larger), this is more conceivable. However, such differences are also evident across within-lab software. For example, Open Sesame (NB. It was not clear to me which back-end was used in the Stage 2 replication by the reviewer) has a much higher variance than the Psychtoolbox, but these differences are small. At another level, the author of the Twitter thread references an attentional blink paradigm. Such a paradigm requires very short presentation durations (1-2 frames), and thus, such timing concerns are also applicable to lab-based testing environments. Indeed, not all in-lab software accurately reports missed flips, and often shows timing parameter differences across operating system/graphic drivers and so on. For an attentional blink paradigm, I would only be comfortable running such a paradigm within a lab-setting running something like the Psychtoolbox under a Linux-based system (rt-kernel) on a CRT monitor with an external photo-diode or Black-Box toolkit attached. But the critical comparisons within the current experiment do not involve very short presentation durations, low-level perceptual manipulations, or the requirement for precise visual-audio synchronization. I do not understand the author’s concern about systematic condition differences and how they could influence the results under certain conditions. And this is sometimes a general problem with “boilerplate/cookie-cutter” responses in reviews: they are not specific to the article under review. Also, I am a little bit unsure how a researcher should respond to such concerns (beyond citing specific studies that have explicitly investigated the reliability of such methods, in this specific case online measures of reaction times).

Appendix C

No changes required following IPA

I now completed the required steps (submitting the Stage 1 report to OSF, analyzing and interpreting the data, etc.)

Appendix D

Associate Editor Comments to Author (Professor Chris Chambers):

Comments to the Author:

The three reviewers who assessed the Stage 1 submission kindly returned to evaluate the Stage 2 manuscript. As you will see, all are positive about the completed article and judge that the primary Stage 1 criteria are met. There are some minor issues to attend to in revision, including the curation of the online data, some minor inconsistencies with the original protocol, correct interpretation of non-significant findings, and presentational issues (e.g. concerning tenses and typos). Regarding Reviewer 3's concern with your statement, "This preregistration was performed after data analysis": since you submitted via the results-blind track, your statement is correct.

Provided you are able to respond comprehensively to all points raised in a minor revision and response, final acceptance should be forthcoming without requiring further in-depth review.

Response: Thanks for the overall positive evaluation. The manuscript was now proof-read in detail. I apologize for the delay. I also checked the analysis again and realized that for the combined analysis of the two experiments an outlier cutoff wasn't correct in the analysis scripts, the adjustment resulted in some minor changes in the decimal places of the statistical values – without any change in significance levels or other relevant aspects.

Reviewers' comments to Author:

Reviewer: 1

Comments to the Author(s)

This study presents a very clear replication attempt of Kan et al., (2013). I am impressed by the openness about the analyses and the details provided. The conclusions drawn seem to be appropriate and based on the data.

I just have two comments. One of the comments is with regard to the data and analyses shared: meta data and explanations of your data/variables will help to understand every column of your data and would make your analyses more reproducible and the data more reusable.

Additionally, there seem to be quite some spelling mistakes in the manuscript.

Response: Thanks for pointing this out, the manuscript was now carefully checked for spelling mistakes. There is now a readme.txt file accompanying the data which should clarify all remaining issues.

Reviewer: 2

Comments to the Author(s)

Summary

This is the stage 2 version of a stage 1 replication report that I had previously reviewed. The manuscript attempts to replicate the findings of Experiment 1 from Kan et al. (2013).

Review

I thought this was a well-conducted replication and I have only a handful of minor comments. The analysis seemed to me to be consistent with the plan (though see minor comment in point

1). The authors additionally removed extremely long RTs before analysis, which was not pre-registered but is entirely sensible to me. The interpretation of the results seems well-justified.

Response: Thanks for sharing this Yes, it was decided that this change to the pre-registration plan seemed more sensible with regard to conclusions that can be drawn from the studies. It is indicated in the manuscript that this step deviated from the pre-registered plan.

1) The pre-registered analysis plan states that the results would focus on raw error rates (pg 12, line 5), but the results state that the focus was on the arcsin transformed error rates (pg 14, line 26). It appears to make little difference – the manuscript states that both would be reported if they differed and I didn't see an instance of it - but it is an inconsistency with the analysis plan.

Response: The results now focus on the arcsin transformed error rates as these were analysed in the original paper (which a reviewer pointed out earlier). The analyses scripts contain all options. There was no theoretical consistent difference between both analyses. I now clarify in the paper that this deviated from the as-predicted pre-registration.

2) Wording on page 12 line 3: “Again, here report both analyses are reported...”

Response: Thanks, now corrected.

3) Wording on page 28 line 12: “...participant sample was use based.”

Response: Thanks, now corrected.

Reviewer: 3

Comments to the Author(s)

I was happy to see that this project reached completion. Although the results are not surprising, I believe that it's important to publish data and results relevant to the investigated question.

I only found minor problems with the submission:

The manuscript has a good number of sentences written in future tense. Searching for “will” or “will be” quickly shows where. Some of these are relevant, but other must be left here from Stage 1.

Response: Thanks, this is now corrected.

p.6.

"This preregistration was performed after data analysis." might be a mistake.

Response: This was correct at this point (see editors comment above). The preregistration on as-predicted was done before data collection.

p.14

In “For RT, the ANOVA revealed a significant main effect of Stroop congruency with faster responses to congruent trials (654 ms)” and similar places, it must be mentioned that these are mean values.

Response: This is now clarified across the manuscript.

P.16

“was not significant, $F(1, 89) = 1.46$, $p = .231$, $\eta^2 = 0.02$, indicating that the congruency effect was similar following compatible trials“ is an incorrect conclusion as non-significant results cannot indicate such conclusion.

Response: Thanks, this is now corrected (“... was not different ...”).

P.20

Please double-check the “difference in both raw and residual reading times (~11 ms difference) within the first critical region, $t_s(94) > 4.03$, $p_s 0.41$ ” results. Appendix table indicates that the SD was 100 ms here, so I was surprised to see such a strong effect for 11ms difference.

Response: This *SD* is not relevant for the significance of the within-participant *t*-test.

Balazs Aczel